# INPUTDSA : DEMIXING THEN COMPARING RECURRENT AND EXTERNALLY DRIVEN DYNAMICS

**Ann Huang**[1,2,3,*]**, Mitchell Ostrow**[4,*]**, Satpreet H. Singh**[2,3]**,**
**Leo Kozachkov**[5]**, Ila Fiete**[4]**, Kanaka Rajan**[2,3]

[1]Harvard University    [2]Harvard Medical School    [3]Kempner Institute
[4]Massachusetts Institute of Technology    [5]Brown University

{annhuang@g.harvard, ostrow@mit}.edu

[*]Equal contribution

## ABSTRACT

In control problems and basic scientific modeling, it is important to compare observations with dynamical simulations. For example, comparing two neural systems can shed light on the nature of emergent computations in the brain and deep neural networks. Recently, Ostrow et al. (2023) introduced Dynamical Similarity Analysis (DSA), a method to measure the similarity of two systems based on their recurrent dynamics rather than geometry or topology. However, DSA does not consider how inputs affect the dynamics, meaning that two similar systems, if driven differently, may be classified as different. Because real-world dynamical systems are rarely autonomous, it is important to account for the effects of input drive. To this end, we introduce a novel metric for comparing both intrinsic (recurrent) and input-driven dynamics, called InputDSA (iDSA). InputDSA extends the DSA framework by estimating and comparing both input and intrinsic dynamic operators using a variant of Dynamic Mode Decomposition with control (DMDc) based on subspace identification. We demonstrate that InputDSA can successfully compare partially observed, input-driven systems from noisy data. We show that when the true inputs are unknown, surrogate inputs can be substituted without a major deterioration in similarity estimates. We apply InputDSA on Recurrent Neural Networks (RNNs) trained with Deep Reinforcement Learning, identifying that high-performing networks are dynamically similar to one another, while low-performing networks are more diverse. Lastly, we apply InputDSA to neural data recorded from rats performing a cognitive task, demonstrating that it identifies a transition from input-driven evidence accumulation to intrinsically-driven decision-making. Our work demonstrates that InputDSA is a robust and efficient method for comparing intrinsic dynamics and the effect of external input on dynamical systems.

## 1 INTRODUCTION

Identifying that two seemingly disparate complex systems have the same underlying structure is a widespread objective across many scientific fields, including deep learning (Huh et al., 2024), computational and systems neuroscience (Yamins et al., 2014; Aldarondo et al., 2024; Prinz et al., 2004), and physics (Hohenberg & Halperin, 1977; Feigenbaum, 1978). One common approach to characterizing the similarity of two systems (e.g., brains, minds, computational models, or physical objects) is to compare the geometry of their states. Well-known methods to do so are Representational Similarity Analysis, Centered Kernel Alignment, Procrustes Analysis, Canonical Correlation Analysis, and Pearson Correlation (Kriegeskorte et al., 2008; Kornblith et al., 2019; Williams et al., 2022; Gallego et al., 2018; Raghu et al., 2017; Schrimpf et al., 2018). Neural networks can also be characterized by the topology of their activations, (Chaudhuri et al., 2019; Gardner et al., 2022; Lin & Kriegeskorte, 2024), a more invariant measure than geometry, which depends on the particular sampling of neurons from the network. However, common to all is that they do not capture similarity in temporal dynamics (Galgali et al., 2023; Maheswaranathan et al., 2019; Ostrow et al., 2023).

Metrics such as Dynamical Similarity Analysis (DSA, Ostrow et al. 2023) offer an important complementary lens to structure characterization, by proposing a similarity metric on the level of dynamics. DSA provides an efficient and theoretically grounded dynamical similarity metric that has been successfully applied to recurrent network dynamics, training dynamics, and biological neural

data (Redman et al., 2024a; Huang et al., 2025; Codol et al., 2024a; Guilhot et al., 2024; Versteeg et al., 2025; Lazzari & Saxena, 2025). Briefly, DSA nonlinearly embeds dynamics into a high-dimensional space and estimates a linear state-transition operator from observed trajectories, which is then compared across systems. Recent work introduced other methods for dynamics comparison (Redman et al., 2024b; Vermani et al., 2024; Cotler et al., 2023; Gosztolai et al., 2025; Chen et al., 2024; Nejatbakhsh et al., 2024) based on other computational techniques such as neural networks and shape metrics. Notably, none of these methods consider the effect of external input.

In neuroscientific settings such as central pattern generators or working memory circuits, dynamics may be treated as approximately autonomous (Marder & Bucher, 2001; Grillner, 2006; Kiehn, 2016; Fuster & Alexander, 1971; Funahashi et al., 1989; Goldman-Rakic, 1995; Compte et al., 2000; Wang, 1999). Prior methods work well for comparisons in these settings. However, when activity is the result of both intrinsic dynamics and input drive, comparisons can be confounded by inputs. Most systems of interest in neuroscience and machine learning are non-autonomous, receiving sensory signals or communication from other subsystems (Eisen et al., 2025). They are driven by complex external inputs and can receive observations that are contingent on the systems' outputs (Madhav & Cowan, 2020; Kao & Hennequin, 2019; Rajan et al., 2010).

Despite the ubiquity of input, current dynamical similarity methods ignore input-driven dynamics and do not incorporate estimation of how inputs affect states. To bridge this gap, we introduce InputDSA (iDSA), a method that disentangles intrinsic dynamics from input-drive, thereby enabling joint or separate metric comparisons of input-driven and intrinsic dynamics. InputDSA extends the DSA framework by explicitly estimating both the intrinsic (state-transition) operator and the input-to-state mapping, which not only defines a new notion of similarity that incorporates the effect of inputs, but also in turn improves estimation of the intrinsic operator.

**Contributions** We extend DSA to non-autonomous systems that are driven by external input, which we call InputDSA . To do so, we develop a novel similarity metric and variant of the dynamic mode decomposition (DMD), demonstrating that they can together provide complementary insights on both intrinsic as well as input-driven dynamical similarity. We demonstrate InputDSA first on systems with known ground truth. We next show that similarity scores can be robust to surrogate or noisy inputs, provided that they have sufficient similarity to the real inputs. Finally, we apply InputDSA to two datasets: RNNs trained with Reinforcement Learning, and neural population data (spiking) from rats performing a sensory decision-making task. We show that InputDSA distinguishes high- from low-performing models and reveals how dynamics reorganize across different task periods.

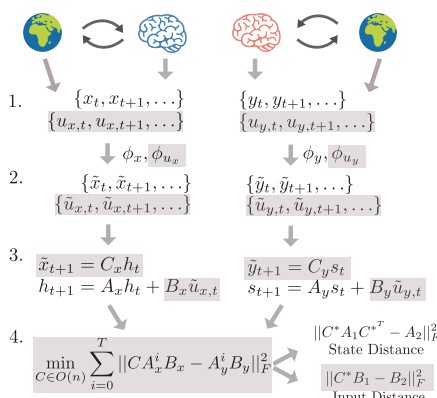

Figure 1: **InputDSA schematic** (1), state and input data are collected from two systems. (2) data are embedded in a high-dimensional space (3) linear state-space models are fit to the data (4) Controllability, state, and input similarity are computed on learned state-space models. Gray indicates extensions from DSA.

## 2 METHODS

### 2.1 DYNAMICAL SIMILARITY ANALYSIS (DSA)

In dynamical systems, a key notion of similarity is called topological conjugacy: the existence of a homeomorphism that maps trajectories of one system onto those of another. When two systems are conjugate, they have the same qualitative structure, including the same number and type of fixed points. Given two dynamical systems $f : X \to X$ and $g : Y \to Y$ with mapping $\phi : X \to Y$, (semi-) conjugacy is defined as:

$$g \circ \phi = \phi \circ f \tag{1}$$

The existence of such a mapping entails a one-to-one alignment between topological features of each system such as invariant manifolds. Note that this is not geometric because distances and angles are not necessarily preserved under this mapping. In general, such a function can be arbitrarily complex,

which can make searching for the true conjugacy map challenging in all but the simplest settings. DSA attempts to circumvent the optimization problem by approximating the Koopman Operator, which linearizes nonlinear dynamical systems via high-dimensional embeddings (Koopman, 1931; Budišić et al., 2012). In the linear space, conjugacy maps are linear and therefore easier to identify. The methodology of DSA is therefore as follows: First, approximate your systems as linear in some high-dimensional space, yielding dynamics models $x_{t+1} = Ax_t$ . Then similarity is defined on the linear operators using the following metrics:

$$\text{DSA}(A_1, A_2) := \min_{C \in O(n)} ||CA_1C^T - A_2||_F \tag{2}$$

$$\text{DSA}(\Lambda_1, \Lambda_2) := \min_{P \in \Pi(n)} ||P\Lambda_1P^T - \Lambda_2||_F \tag{3}$$

Where $\Lambda_i$ is the eigenvalue matrix of $A_i$, and $O(n), \Pi(n)$ the groups of $n \times n$ dimensional orthogonal and permutation matrices. The latter metric was introduced by (Redman et al., 2024b) and is a special case of the former (Ostrow et al., 2023). These metrics are reminiscent of Procrustes Analysis, which seeks an orthogonal transformation to align two data matrices, hence Ostrow et al. (2023) termed the first one Procrustes Analysis over Vector Fields. The latter metric is inspired from Koopman Operator Theory based on the relationships between Koopman Operators of conjugate systems (Budišić et al., 2012). Other notions of similarity on the Koopman Operator are defined in (Mezić & Banaszuk, 2004; Mezic, 2016).

## 2.2 INPUTDSA

Inspired by DSA's approach for autonomous systems, consider two *linear* dynamical systems

$$\dot{x} = A_1x + B_1u(t) \qquad \dot{y} = A_2y + B_2u(t). \tag{4}$$

A key feature of input-driven systems is their *controllability*: the ability for an input sequence to drive the state to arbitrary points in finite time. In linear systems, this is encoded in the T-step controllability matrix (with T typically taken as the dimension of the system):

$$K_1(T) = \begin{pmatrix} B_1 & A_1B_1 & A_1^2B_1 & \dots & A_1^{T-1}B_1 \end{pmatrix} \tag{5}$$

and its corresponding Gramian, which encodes the geometry of controllability.

$$W_c(T) = K_1K_1^T = \sum_{i=0}^{T} A_1^iB_1(A_1^iB_1)^T \tag{6}$$

Intuitively, directions with small eigenvalues are easier to control, because they are more responsive to the effect of input. Controllability, as measured by the eigenvalues of the Gramian, is only preserved under orthogonal transformations between state spaces (Appendix G.2):

$$y = Cx \implies A_1 = CA_2C^T, B_1 = CB_2 \quad K_1 = CK_2 \tag{7}$$

This motivates our proposed dissimilarity metric, which extends DSA:

$$\text{InputDSA}\ (A_1, A_2, B_1, B_2, T) = \min_{C \in O(n)} \sum_{i=0}^{T} ||CA_1^iB_1 - A_2^iB_2||_F^2 = \min_{C \in O(n)} ||CK_1 - K_2||_F^2 \tag{8}$$

We also provide a theoretical extension of Eq. 3 in Appendix G.1, which we note is highly susceptible to numerical instability. Although Eq. 2 requires iterative optimization, Eq. 8 is solved via Procrustes alignment, which yields an *exponential* acceleration of prior work. We provide theoretical discussion in Appendix G. After solving for $C^*$, we can study the joint state and input scores:

$$\text{InputDSA}\ _{\text{state}}(A_1, A_2, C^*) = ||C^*A_1C^{*T} - A_2||_F^2 \tag{9}$$

$$\text{InputDSA}\ _{\text{input}}(B_1, B_2, C^*) = ||C^*B_1 - B_2||_F^2 \tag{10}$$

If the inputs directly applied to the system are known, as in computational models, Eq. 8 is sufficient. However, when the true input is some modification of a surrogate input, it may be necessary to align the input as well. This is relevant in settings such as the comparison of two brain regions, when the surrogate input $u$ is a behavioral or sensory variable that is transformed by upstream regions. We therefore can extend Eq. 8 to consider joint alignment of the input, without significant differences in the optimization problem. For further technical details, see Appendix F.

This metric motivates the following approach as in Ostrow et al. (2023): identify the best linear approximation of an input-driven system, following which comparison can be done efficiently between the approximations. To do so, Ostrow et al. (2023) applied the Dynamic Mode Decomposition (Schmid, 2022), which we introduce and extend to fit our setting next.

## 2.3 ESTIMATING LINEAR OPERATORS

As in DSA, We fit linear operators via the Dynamic Mode Decomposition (DMD) family of methods. The DMD (Schmid, 2010; 2022) identifies the linear dynamics that best explain the data:

$$\phi(x_{t+1}) = A\phi(x_t). \tag{11}$$

Here $x_t$ represents the measured state of the system at time $t$, $\phi$ is a nonlinear embedding of the data that typically expands the dimensionality of the state space, and $A$ is a matrix that is identified using some variant of least-squares regression. The goal of the Dynamic Mode Decomposition is to approximate the Koopman Operator (Koopman, 1931), a theoretical object that exists for all dynamical systems which encodes the linear dynamics of observables (functions that act on the true system state) under the system dynamics. Prior work has explored many different choices of $\phi$. For example, $\phi$ can be a kernel function, a delay embedding, or even a neural network (Williams et al., 2016; Brunton et al., 2017; Arbabi & Mezić, 2017; Lusch et al., 2018). Intuitively, the dimensionality expansion acts similarly to the kernel trick (Smola & Schölkopf, 1998), where embedding into higher dimensions 'unfolds' the nonlinearity. The DMD can be applied in non-autonomous systems, although this risks mixing driving and intrinsic dynamics (Proctor et al., 2016a).

**Incorporating Control into DMD and Koopman**  While the original Koopman theory assumed autonomous dynamics, prior work has sought to incorporate control into the theory (Korda & Mezić, 2018; Proctor et al., 2016b; Strässer et al., 2025; Asada & Solano-Castellanos, 2024; Haseli et al., 2025). Likewise, the DMD can be generalized to driven systems: When given control inputs $u_t$, we can instead apply DMD with control (DMDc, Proctor et al. 2016a;b):

$$\phi_1(x_{t+1}) = A\phi_1(x_t) + B\phi_2(u_t) \tag{12}$$

Here, $\phi_1$ and $\phi_2$ can be distinct nonlinearities. While DMDc was originally only applied with no nonlinearity ($\phi_1, \phi_2 = \text{Id}$), it too can be generalized to high-dimensional nonlinear embeddings. As in DMDc, we assume that inputs are known or a useful surrogate can be constructed (Section 3.2). Further algorithmic details are in Appendix B.

**Issues of Partial Observation**  While estimating $A$ and $B$ via DMDc is an intuitive extension to input-driven systems, it has a hidden failure mode in the analysis of partially-observed systems. This is particularly important in the analysis of neural data, in which a small subset of neurons in a vast population are recorded. Generically, an input-driven system that is partially observed receives input to both the observed and unobserved components. The input at time $t$ therefore affects the observed state at time $t$ (instantaneously) and in future time steps through the unobserved state (Fig. 2A). This means that simply applying DMDc in this setting will bias the $B$ matrix towards the intrinsic dynamics of the system. We develop a formal description of this problem for linear systems in Appendix D. We solve this problem by introducing **Subspace DMDc**, an extension of Subspace DMD (Takeishi et al., 2017b) that incorporates input. In brief, Subspace DMDc utilizes subspace identification algorithms from classical control theory (Verhaegen & Verdult, 2007), which seek to identify linear dynamical systems of the form:

$$x_{t+1} = Ax_t + Bu_t \qquad y_t = Cx_t \tag{13}$$

Here, only $y_t$ and $u_t$ are observed. The situation of partial observability is a special case of this problem. In practice, we use the well-known N4SID or PO-MOESP algorithms to estimate $A$ and $B$ (Van Overschee & De Moor, 1994; Verhaegen, 1994) in Eq. 13 on lifted states (thereby leveraging the power of nonlinear DMD algorithms such as Williams et al. 2016). The algorithmic idea behind subspace identification is similar to that of instrumental variable regression: the lifted state data to-be-predicted (future data $y_{t+1}$) is projected onto the basis of the past input and lifted state data ($y_t, u_t$) before estimating a $A$ and $B$ via linear regression. This has the added benefit of projecting out observation and process noise, thereby providing noise robustness (Verhaegen, 1994; Verhaegen & Verdult, 2007). For technical details on the subspace identification algorithm, see Appendix E.

**Tuning SubspaceDMDc for InputDSA**  SubspaceDMDc has three key hyperparameters: the rank of the linear operator, the number of delays used in subspace identification, and the type of nonlinear basis used for linearizing the dynamics. In practice, the most general method we use to tune SubspaceDMDc is to pick the smallest rank model that best predicts future states of the

system. Because SubspaceDMDc must infer the latent state from prior data, we utilize Kalman Filtering (Kalman, 1963) for efficient next-step prediction. This is effectively done with the Akaike Information Criterion (AIC) along with other metrics we discuss in Appendix I.

## 3 EXPERIMENTS

### 3.1 INPUTDSA DISCRIMINATES INTRINSIC DYNAMICS FROM INPUT-DRIVEN DYNAMICS

To demonstrate that InputDSA can capture similarities in both intrinsic and input-driven dynamics, we simulated partially observed random RNNs with the following equations:

$$x_{t+1} = A(x_t + gF \tanh(x_t)) + B(u_t + \tanh(u_t)) \tag{14}$$

$$y_t = (\mathbf{I}_d \quad \mathbf{0}_{n-d}) x_t + \epsilon_t \tag{15}$$

Where F and g are fixed across all simulations, and $\epsilon_t$ is observation noise. We randomly sampled two matrices for $A \in \mathbb{R}^{n \times n}$, and two for $B \in \mathbb{R}^{n \times 1}$, from which we constructed 4 systems: Systems 1 and 2 (3 and 4) share the same intrinsic dynamics matrix $A_1$ ($A_2$), while Systems 1 and 3 (2 and 4) share the same input matrix $B_1$ ($B_2$). We randomly sampled low-pass filtered white noise as the input drive (four times for each system), each with random initial conditions, yielding 16 systems in each distance matrix. In our experiments, we simulated 20-dimensional ($x \in \mathbb{R}^{20}$) systems and observed 2 dimensions ($y \in \mathbb{R}^2$) for 5,000 time points. For simulation details, see Appendix J. We computed 5 distance matrices for each dataset, across 100 random seeds: (1), the DSA score using a delay-embedded DMD (Hankel DMD, or Hankel Alternative View of Koopman Arbabi & Mezić 2017; Brunton et al. 2017), (2) the state distance using a delay-embedded DMDc, (3) the state distance using the SubspaceDMDc, (4) the input distance using the DMDc, and (5) the input distance using the SubspaceDMDc. Note that DSA does not have the ability to compare inputs, so it is left out. For a discussion on hyperparameter tuning, see Appendix J). For the sake of space, we report the jointly optimized input distance (Input DSA, Eq. 10) and the individually optimized state distance (State DSA, Eq. 9) as these are the most interpretable, although the jointly optimized state distance was highly similar.

In Fig. 2B we visualize the observed input and one dimension of the observed output for a sample set of systems, noting that it is not obvious at all a priori, let alone from the geometry, of any similarity relationships. We present sample state distance matrices from one random seed in Fig. 2C. While the DMD and the DMDc have notable structure pertaining to the true state similarity, the SubspaceDMDc similarity scores are noticeably sharper. Quantifying these matrices with the silhouette score (a measure of cluster separability and dispersal, 1.0 is best) utilizing ground-truth state labels, the DMD scores 0.6, DMDc scores 0.68, and the Subspace DMDc scores 0.94. In Fig. 2D, we present the respective input scores for each method. As predicted by our previous discussion on the effects of partial observation on input matrix estimation, the input DSA score computed with DMDc does not align with ground truth, reporting a silhouette score of 0.19. The silhouette score of the SubspaceDMDc is 0.83, indicating robust separation. We also compute the total similarity matrices (Eq. 8, Appendix Fig. 6), for which the SubspaceDMDc reports correctly that each type of system is altogether unique. We swept over 100 seeds in Fig. 2E and found that the SubspaceDMDc-based InputDSA consistently yielded the best separability.

To assess the effect of partial observation, we ran the above analysis for different-sized systems (ranging from 2 to 1000 dimensions) with only 2 observed dimensions, for which we present the average silhouette scores for InputDSA in Fig. 2F. The state similarity scores for each method gracefully degrade with the total state size, and SubspaceDMDc has a noticeable improvement over the other methods. The DMDc input score appears to never be robust. However, the SubspaceDMDc input similarity is robust across all system sizes. We also assess the effect of hyperparameters, input dimensionality, and process noise on SubspaceDMDc (Appendix Figs. 8, 9, 10 respectively). Together, this suggests that SubspaceDMDc can be used to measure the dynamical similarity of partially observed, input-driven, noisy dynamical systems.

### 3.2 ROBUSTNESS TO INPUT NOISE AND TRANSFORMATION

In real-world settings such as neural populations, the true inputs driving the system are rarely accessible. Instead, what we observe are often noisy or partial measurements, limited by sensor resolution,

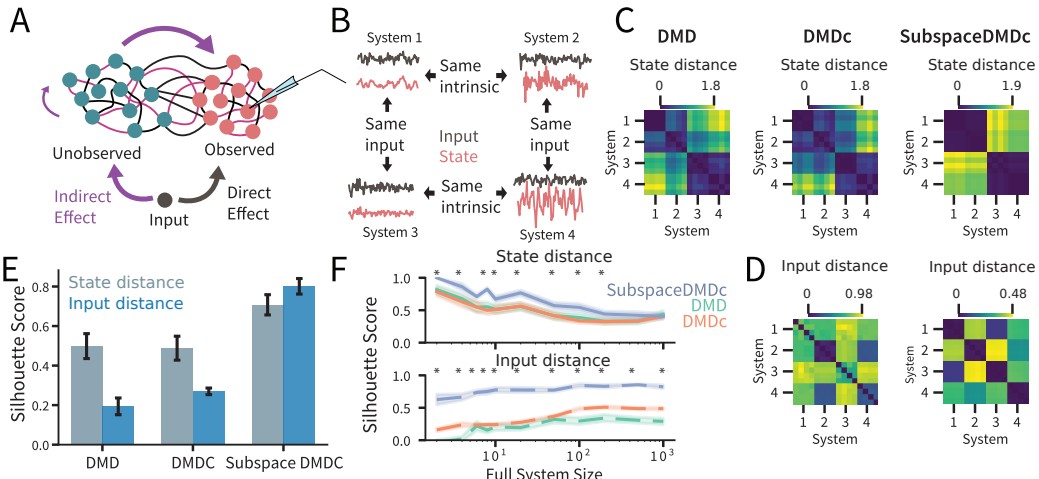

Figure 2: **InputDSA SubspaceDMDc is robust to partial observation** **(A)** Inputs can affect observed states (red nodes) in the future via the unobserved states (green nodes), biasing estimates of input driven-dynamics. Purple arrows indicate indirect propagation of input. **(B)** Sample inputs and observed states from 4 dynamical systems, which have alternate pairings of the same intrinsic and input-driven dynamics denoted by arrows. **(C)** Sample state distance matrices based on the DMD, DMDc and SubspaceDMDc on data generated as in (B). Four iterations of each system are generated, each with unique inputs and initial conditions. **(D)** Sample input distance matrices on the same data as in (C). The DMD does not learn an input operator. **(E)** Aggregate silhouette scores of each similarity matrix across 100 random seeds, each generated as in (C,D). Baseline input-label silhouette score for DMD is computed on the state matrix with the ground-truth input labels. Bars denote standard error. **(F)** Silhouette scores for each DMD and similarity type as the system is increased from 2- to 1000-dimensional. Each size was repeated across 20 seeds. Shading denotes standard error, asterisks indicate that SubspaceDMDc scores are significantly greater than both DMD and DMDc (one-side Mann-Whitney U-Test).

sampling rates, or inherent partial observability. As a consequence, researchers often rely on behavioral variables, task instructions, or environmental features as proxies when modeling neural circuits (Vinograd et al., 2024; Sani et al., 2024; Burak & Fiete, 2009b; Schaeffer et al., 2020; Mante et al., 2013b). This raises a key question for applying InputDSA : if the true inputs are unknown, can *surrogate inputs* that are correlated with the ground truth still yield accurate distance estimates?

We begin by examining how well the true InputDSA distance matrix when the provided input is noise-corrupted. We repeated the simulation and comparison in Fig. 2C, this time applying different types of noise perturbations to the input used in SubspaceDMDc (example in 3A). For complete details on the noise perturbations, see Appendix M. We applied 10 types of perturbations inspired by different real-world situations, such as partial observation, temporal smoothing, or multiplicative Gaussian noise, and repeated each perturbation across a range of parameters (e.g. standard deviation in the noise settings or filter width in the smoothing setting).

To measure the deviation of the signal consistently across perturbation types, we compute the signal-to-error ratio (SER) for each perturbation: given a time series $X \in \mathbb{R}^{t \times d}$ and its perturbed version $\tilde{X} = f(X)$, SER is defined as $SER(X, \tilde{X}) = \frac{\text{Var}(X)}{\text{Var}(\tilde{X} - X)}$. SER generalizes signal-to-noise ratio for non-additive perturbations. Despite the prevalence of noisy inputs, we found that InputDSA distances remain robust, decaying slowly below the $SER < 1$ threshold (Fig. 3B): High SERs lead to high correlations with ground truth distances, and correlations tend remain above $r > 0.75$ even as SER approaches 1. This robustness arises due to the delay embedding and reduced-rank regression in Subspace DMDc: delay embedding incorporates the history of inputs, while reduced-rank regression removes noisy modes with spurious correlations.

To generalize this analysis to more complicated transformations, we repeated the analysis using inputs transformed by random polynomials (Fig. 3C,D). Specifically, we sampled 500 random 4-th order polynomials with coefficients drawn uniformly from $[-0.1, 0.1]$, which we applied dimension-wise to the inputs as a new perturbation. To generate inputs with higher SERs, we also generated 200 polynomials where the linear coefficient was fixed at $0.9$, while all other coefficients were sampled

from the same range. As in the previous analysis, we find a similar pattern across SER: the state DSA correlations are the most robust, followed by the combined and the input DSA scores. This suggests that up to reasonable SER , the InputDSA scores are robust to input perturbations.

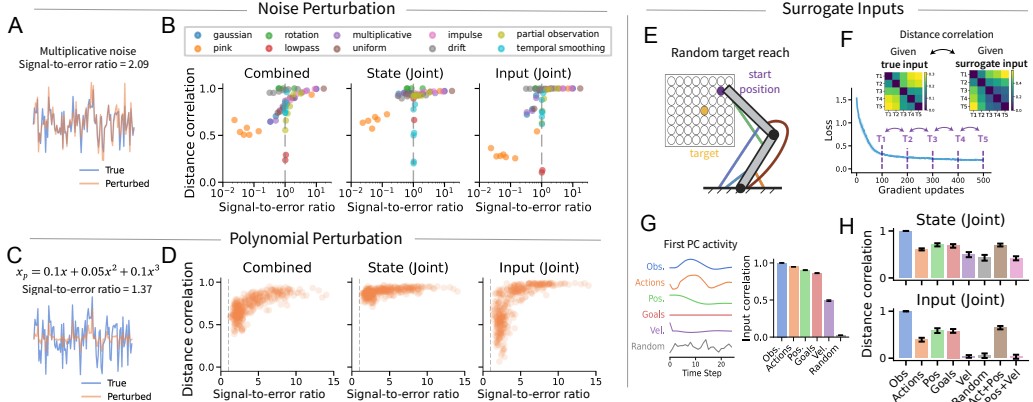

Figure 3: **InputDSA provides robust distance estimates under input noise and surrogate inputs.** **(A)** Example of multiplicative Gaussian noise added to input data. **(B)** Effect of different noise perturbations on the InputDSA similarity matrices in Fig. 2 (see Appendix Section M for further technical details on the noise). The y-axis indicates the correlation between the InputDSA matrices given the true input and the perturbed input. The x-axis indicates the signal to error ratio $\text{Var}(X)/\text{Var}(\tilde{X} - X)$. From left to right: joint controllability DSA (Eq. 8), jointly optimized state DSA (Eq. 9; jointly optimized input DSA (Eq. 10). **(C)** Example of a polynomial function applied to the same input as in (A). **(D)** Similar analysis as in (B), with various random polynomial functions applied to the input. **(E)** Random target task schematic. **(F)** We compare RNNs dynamics across multiple time points in training with InputDSA . We study changes in the distance matrix when applying surrogate inputs. **(G)** Example first Principal Component for different surrogate inputs and their correlation with the true input (Obs). **(H)** Correlation between InputDSA distances estimated using the ground truth input and surrogate inputs. Error bars indicate standard error across 10 training runs. Jointly optimized state and input DSA are presented.

Next, we evaluated whether task-relevant surrogate inputs could be used in place of ground truth, instead of perturbed versions of the true input. We analyzed trained RNNs from the Random Target Reach task (Fig. 3E, Codol et al. 2024b), a widely used paradigm for studying neural control of movement from which rich neural and behavioral dynamics emerge (Hatsopoulos et al., 2007; Flint et al., 2012; Churchland et al., 2012). Across 20 epochs equally spaced in training, we recorded the RNN's hidden states, observations (the true input), actions (behavioral output), and other task variables (Fig. 3F, only 5 epochs shown for visualization purpose). For a detailed description of the task and training, see Appendix O. Passing the hidden states of the RNN and the ground truth inputs through InputDSA , we obtained two distance matrices that characterize how the network's intrinsic and input-driven dynamics change over learning. We repeated this process for various task-related surrogate inputs: RNN output (actions), position, velocity, task instruction, and various combinations. We also included random inputs sampled from the uniform distribution on $[0, 1]$ as a baseline. Among surrogates, the actions maintain the highest trial-averaged correlation with the ground truth input (Fig. 3G). We find that InputDSA intrinsic (state) distances estimated with surrogate inputs have strong correlation with the ground truth distance, even with random inputs (Fig. 3H). For input-driven comparisons, more highly-correlated surrogates tend to yield more accurate similarities, with the RNN's combined action and positions providing strong correlations with the ground truth distance (Fig. 3H). Overall, our analysis suggests that state similarities are robust to perturbations of many different types, while the combined and input similarities are still robust, albeit less so.

## 4 APPLICATIONS

### 4.1 INPUTDSA TRACKS THE EVOLUTION OF INDIVIDUAL DIFFERENCE OVER LEARNING

In closed-loop Reinforcement Learning (RL) environments, stochastic action selection and small differences in policies can shift the distribution of sensory inputs encountered across training. To understand divergence between agents, it is crucial to study how inputs interact with dynamics and

shape agent performance. The Plume Tracking task (Fig. 4A) provides an ideal testbed because the agents must balance between memory-based intrinsic dynamics with stimulus-driven responses.

In this task, artificial flies (RNNs) trained by deep RL navigate to the source of a simulated turbulent odor plume in a windy 2D arena. At each timestep, the agent senses only local cues (intermittent odor concentration and wind direction) and takes actions to move its position. Due to the stochastic nature of sensory observations and exploration, agents diverge across training, producing a wide variation of success rates (Fig. 3B). This raises a key question: do performance differences reflect variations in intrinsic dynamics (the ability to form and maintain task-relevant representations) or input-driven responses to stimuli?

We trained 15 independent agents on the Plume Tracking task. We selected the five best-performing ("Top") agents with 65% to 20% success rate at locating the odor source across 200 evaluation episodes, and five worst-performing ("Bottom") agents who never succeeded on any episode (Fig. 4B). Applying InputDSA revealed that the input-driven dynamics of the Top agents were significantly more similar to each other and clearly separated from those of the Bottom agents, whereas intrinsic dynamics were not significantly different between groups (Fig. 4D). This suggests that successful plume tracking heavily depends rapid input-driven responses to wind direction and odor concentration. To probe how the input-driven dynamics differ between Top and Bottom agents, we examined the singular values of the input–mapping $B$ in Fig. 4E. Singular values of the operator quantify how strongly input directions are injected into RNN state space.

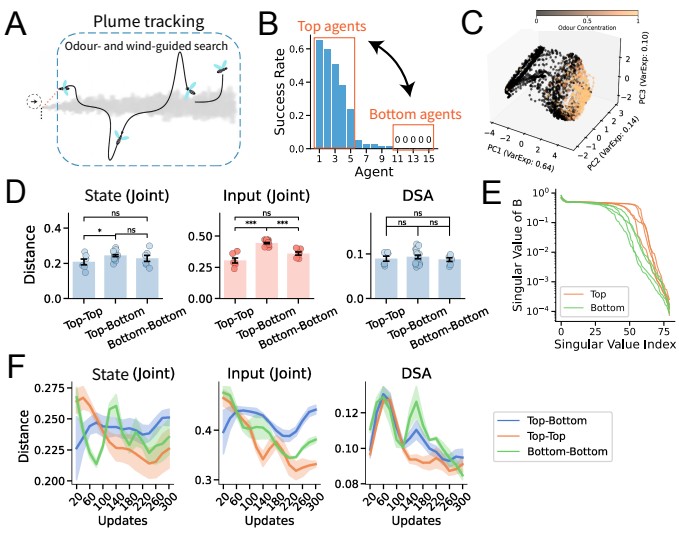

Figure 4: **InputDSA identifies how successful and unsuccessful agents differ over training.** **(A)** The Plume Tracking environment schematic adapted from Singh et al. (2023). **(B)** Average performance (success rate) of 15 independently trained agents. The 5 most performant ("Top") and 5 failed ("Bottom") agents are studied further. **(C)** Neural dynamics of trained agents are organized in a low-dimensional space and reflective of behaviorally relevant variable (i.e. the odor concentration). **(D)** Average distance computed within the 5 Top Agents, within the 5 Bottom agents, and across groups (Top–Bottom). **(E)** The singular value spectrum of the input-mapping operator $B$ from Top and Bottom agents. **(F)** The evolution of similarity within and across groups over learning. Shaded area indicates standard error.

We found that the singular values of $B$ for Top agents decay more slowly than for Bottom agents. This implies that inputs excite more dimensions of the RNN in Top agents (Fig. 4E), allowing them to directly control each dimension with the input, rather than indirectly through other dimensions.

We next ask how individual variability in neural dynamics evolves during training. To this end, we computed pairwise dynamical similarity among Top and Bottom agents every 20 gradient updates (Fig. 4F). While within–group input similarity decreases over training for both Top and Bottom, the Top agents ultimately converge to a more consistent set of input-driven dynamics, whereas the Bottom agents diverge toward heterogeneous, idiosyncratic dynamics. This is reminiscent of the "Anna Karenina principle", in which effective solutions are similar to each other, while worse ones are highly varied. We repeated this comparison using DMDc, finding that it identified similar structure in the input-driven dynamics but not the intrinsic dynamics (Appendix Fig. 17).

## 4.2 INPUTDSA CAPTURES DIFFERENCES IN NEURAL POPULATION DYNAMICS ACROSS TIME

Lastly, we apply InputDSA to a recently published dataset in which neural population activities were recorded from six frontal and striatal regions with Neuropixels probes during an auditory evidence

accumulation task (Luo et al. 2025, Fig. 5A). During this task, rats were trained to listen to auditory pulses from speakers on the either side of the animal, and to turn to the side with more auditory pulses. This dataset contains 12 rats across 115 daily sessions with a median of 327 neurons recorded and 455 trials completed per session. We chose 4 rats with more than 15 recorded sessions for our analysis to ensure accurate estimation of neural dynamics. In the original study, the authors define the *neural time of commitment (nTc)* as the internal moment during perceptual decision-making when an animal has effectively committed to a choice (Fig. 5B). To examine how neural population dynamics reorganize across this point, we applied InputDSA to neural activities during the stimulus presentation period before and after the nTc. Spiking activity was binned in 50 ms windows, smoothed with a causal Gaussian kernel ($\sigma = 250$ ms), and dimensionality reduced with PCA to preserve 99% of variance. The activity was then embedded into three dimensions using Isomap, and InputDSA was applied with hyperparameters detailed in Appendix Q. We construct the inputs as two-dimensional time series encoding the number of auditory pulses from the left and right within each bin. An analysis of the input revealed that this input structure was sufficient to robustly identify a $B$ operator (i.e it is persistently exciting, Appendix Fig. 21).

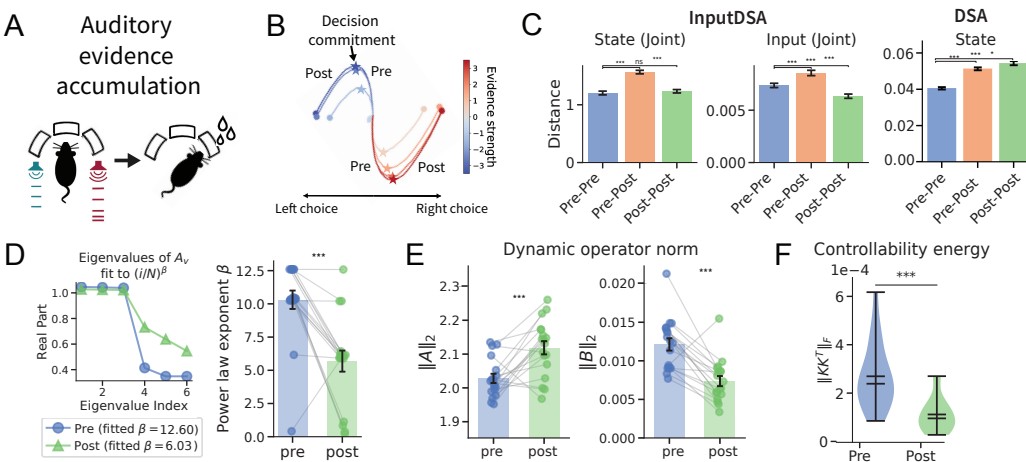

Figure 5: **InputDSA quantifies differences in neural population dynamics across task epoch.** **(A)** Auditory evidence accumulation task schematic (adapted from Luo et al. 2025). **(B)** Trial-averaged neural trajectories visualized in the top two Principal Components. Stars indicate a "neural time of commitment" (*nTc*): the time point when the curvature of trial-averaged trajectories is maximum (marked by stars). **(C)** Similarity of neural dynamics before and after the nTc for rat T223. Bars denote standard error across 21 sessions. **(D)** Distribution of top real eigenvalues of state-transition matrix $A$ and fit power law for pre- vs. post-commitment activity. Left, sample distribution. Right, distribution of power law exponents across sessions. Dots denote individual sessions, lines indicate paired periods within session, likewise in E and F. **(E)** Effects of intrinsic and input-driven dynamics in pre vs. post periods, measured by L2-norm of the SubspaceDMDc operator. **(F)** Distribution of Frobenius norms of Controllability Gramians for SubspaceDMDc models for all sessions of one rat. $p < 0.001$, Mann Whitney U-Test.

Comparing neural dynamics before and after the *nTc* ("Pre" vs. "Post"), we found significant shifts in both intrinsic and input-driven dynamics, consistent with the changes at *nTc* reported in (Luo et al., 2025) (Fig. 5C). To probe how the intrinsic dynamics change, we analyzed the eigenspectrum of the state-transition matrix $A$ estimated by SubspaceDMDc before and after the *nTc*. Each spectrum was fit with a power law $\lambda_i \propto \left(\frac{i}{N}\right)^{\beta}$, where $\lambda_i$ are the $i$-th eigenvalue sorted in descending order. We found that the post-commitment periods consistently showed smaller $\beta$, indicating slower decay and thus longer-lasting intrinsic dynamics (Fig. 5D). This is directly related to the *controllability* of the dynamical system, which describes how easy it is for an input sequence to drive the system to arbitrary points in state space (Luenberger, 1979). Smaller DMD eigenvalues implies greater input controllability, which would be expected for a more input-driven system as Luo et al. (2025) identified is the case in the pre-*nTc* regime. Likewise, the average magnitude of the intrinsic dynamics strengthen while the input-driven dynamics weaken in the Post-nTc period, reflecting a transition into more autonomous, less input-sensitive regime after the *nTc* (Fig. 5E).

Lastly, we computed the Frobenius norms of the controllability Gramians for each SubspaceDMDc model, which measures how easy it is to control the system into arbitrary directions. In Fig. 5F, we report the distribution of norms for one rat pre- and post-nTc. The distribution significantly decays ($p < 0.001$, Mann-Whitney U-test), indicating that the neural dynamics become less input-controllable over time. We found similar results for all other animals in the dataset (Appendix Fig. 22). The median percent change in controllability energy was -47.63% (standard error 15.15%).Together, these results suggest that population activity undergoes a regime shift at the *nTc*: transitioning from an input-driven, evidence-accumulation phase into an intrinsically dominated, decision-commitment phase, as suggested by (Luo et al., 2025). Applying the DMDc for comparison did not reveal similar results, almost certainly due to partial observation (Fig. 19).

## 5  DISCUSSION

We introduced a theoretically-motivated method (InputDSA ) to quantitatively compare the intrinsic dynamics and effects of input between two dynamical systems, from data alone. We extended the DSA framework (Ostrow et al., 2023) to account for input-driven systems, which required a novel variant of the Dynamic Mode Decomposition with Control (Proctor et al., 2016a) called Subspace DMDc. We also developed a novel optimization algorithm for our similarity metric that is multiple orders of magnitude faster than prior work.

We demonstrated that InputDSA can effectively estimate similarity from partially-observed systems (Fig. 2), which is necessary when dealing with most physical and biological systems. In many settings, the true input is not known (for example the signal from one brain region to another), but we demonstrated that even approximate or noisy inputs can provide reasonable input and intrinsic similarity estimates (Fig. 3). Since many models in computational neuroscience tend to utilize proxy inputs (Nair et al., 2023; Sohn et al., 2019; Burak & Fiete, 2009a; Mante et al., 2013a; Sussillo et al., 2015), our work provides principled methodological support to this practice. Inputs could also be estimated via another computational method (e.g. Perich et al. 2020; Luo et al. 2025) before applying InputDSA . As Fig. 3 suggests, even utilizing weakly correlated proxy inputs can increase the robustness of the intrinsic comparison with InputDSA .

InputDSA could be used for further validation of computational models with perturbation as in O'Shea et al. (2022). Known optogenetic or electrical impulse perturbations could be applied to both a model and biological neural circuit, following which both their internal dynamics and impulse responses could be compared. This can provide more stringent tests than comparing intrinsic dynamics alone. Other subspace identification methods could be used in place of SubspaceDMDc, such as Eigensystem Realization (ERA, Juang & Pappa 1985). In a similar vein, InputDSA could potentially be used to identify the information content in cross-brain-region communication – multiple models could be constructed with different surrogate inputs, and the most similar input should have the lowest input distance to the data (Fig. 3).

Although we only applied InputDSA to biological neural data and recurrent neural networks, it can be applied to any time series data. Indeed, the constraints on the method are based on the capabilities of systems identification and Koopman Operator approximation. For example, if the input is not persistently exciting, state modes will be under-approximated. If a viable basis is not identified, the linear model may not be able to capture enough structure for effective comparison. However, there exists a wide range of work in both fields designed to tackle these problems (Wu et al., 2021; Colbrook et al., 2023; Takeishi et al., 2017a; Ichinaga et al., 2024). It is also worth noting that near-perfect estimation is not necessary for informative comparison.

InputDSA has other limitations. The method assumes additive input, which may not be able to approximate the effects of multiplicative input (Logiaco et al., 2021; Shine et al., 2021). Disentangling the contribution of state and input can also be challenging or intractable when they are synchronized (Rajan et al., 2010) or the input is a linear function of the state (Verhaegen & Verdult 2007, although methods exist for subspace identification in closed loop Van Der Veen et al. 2013). From a computational complexity standpoint, the bottleneck is fitting the SubspaceDMDc, as comparison is extremely fast. Regardless, we found that even for reasonably sized systems (e.g. 50 dimensions, 10,000 timepoints) and hyperparameters (100 delays), the method requires a O(1 minute) on M1 Pro Mac, and is even faster on a GPU.

## ACKNOWLEDGEMENTS

We thank members of the Rajan and Fiete labs for helpful discussions. Funded by NIH (RF1DA056403 to K.R.), James S. McDonnell Foundation (220020466 to K.R.), Simons Foundation (Pilot Extension-00003332-02 to K.R.), McKnight Endowment Fund (K.R.), CIFAR Azrieli Global Scholar Program (K.R.), NSF (2046583 to K.R.), Harvard Medical School Neurobiology Lefler Small Grant Award (K.R.), Harvard Medical School Dean's Innovation Award (K.R. and S.H.S.), and Alice and Joseph Brooks Fund Postdoctoral Fellowship (S.H.S.). A.H is supported by the Kempner Graduate Fellowship. M.O. is funded by the NSF GRFP.

## REPRODUCIBILITY STATEMENT

The description of all algorithms utilized in our paper is sufficiently detailed in order to reproduce InputDSA (pseudocode, mathematical details, hyperparameter tuning) and our results. We built upon publically-available code from Ostrow et al. (2023) to create InputDSA . For our RNN studies, we used open source code, and have detailed the hyperparameters we used in the simulations as well as in our analyses. For our neural data analysis, we used public data and processed it generically, thereby making reproduction easy, if desirable. We are thankful for prior open source work and we will release our method open source upon acceptance. We hope InputDSA is not only used and but also improved upon.

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

# Appendix

## OUTLINE

## A    LLM USAGE STATEMENT

We used LLMs in preliminary phases of conducting this research, in particular for brainstorming research ideas and literature review, as well as writing simple boilerplate code (e.g. plotting). All code, math, and writing was checked by at least one author before including it in the paper.

## B    DYNAMIC MODE DECOMPOSITION WITH CONTROL (DMDC)

Dynamic Mode Decomposition with control (DMDc) (Proctor et al., 2016a) extends standard DMD to dynamical systems with external inputs. It provides a data-driven approximation of both the intrinsic dynamics $A$ and input couplings $B$, enabling system identification and forecasting for non-autonomous dynamical systems. Here, we briefly review the formulation of DMDc. For full details, please refer to Proctor et al. (2016a). In practice, we can apply DMDc whenever the state is fully observed. When this is not the case, refer to Sections D and E.

We consider the input-driven linear model

$$x_{k+1} = Ax_k + Bu_k, \qquad A \in \mathbb{R}^{n \times n}, \ B \in \mathbb{R}^{n \times p}, \tag{16}$$

where $x_k \in \mathbb{R}^n$ are state snapshots and $u_k \in \mathbb{R}^p$ are input signals. For an input-driven dynamical system, we collect pairs of the system states and input signals into

$$X = \begin{bmatrix} x_1 & x_2 & \cdots & x_{m-1} \end{bmatrix}, \tag{17}$$
$$X' = \begin{bmatrix} x_2 & x_3 & \cdots & x_m \end{bmatrix}, \tag{18}$$
$$U = \begin{bmatrix} u_1 & u_2 & \cdots & u_{m-1} \end{bmatrix}, \tag{19}$$

where $X, X' \in \mathbb{R}^{n \times (m-1)}$ and $U \in \mathbb{R}^{p \times (m-1)}$. We can rewrite equation 16 into

$$X' = G\Omega = \begin{bmatrix} A & B \end{bmatrix} \begin{bmatrix} X \\ U \end{bmatrix} \tag{20}$$

where $\Omega \in \mathbb{R}^{(n+p) \times (m-1)}$ and $G \in \mathbb{R}^{n \times (n+p)}$.

The optimal operator is then obtained by solving

$$G = \arg\min_{\tilde{G}} \|X' - \tilde{G}\Omega\|_F = X'\Omega^+, \tag{21}$$

where $(\cdot)^+$ denotes the Moore–Penrose pseudoinverse.

Let the truncated SVD of $\Omega$ be

$$\Omega \approx \tilde{U}\,\tilde{\Sigma}\,\tilde{V}^*, \qquad \Omega^+ \approx \tilde{V}\,\tilde{\Sigma}^{-1}\tilde{U}^*. \tag{22}$$

Partition $\tilde{U}$ into state and input blocks:

$$\tilde{U} = \begin{bmatrix} \tilde{U}_x \\ \tilde{U}_u \end{bmatrix}, \qquad \tilde{U}_x \in \mathbb{R}^{n \times \tilde{r}}, \ \tilde{U}_u \in \mathbb{R}^{p \times \tilde{r}}. \tag{23}$$

The system matrices are then estimated as

$$A = X'\tilde{V}\tilde{\Sigma}^{-1}\tilde{U}_x^*, \qquad B = X'\tilde{V}\tilde{\Sigma}^{-1}\tilde{U}_u^*. \tag{24}$$

We can further project $A$ and $B$ into the system's state space using

$$X \approx U_r\Sigma_r V_r^*, \qquad U_r \in \mathbb{R}^{n \times r}, \tag{25}$$

$$\tilde{A} = U_r^* A U_r, \qquad \tilde{B} = U_r^* B. \tag{26}$$

It is useful to perform SVD independently on $X$ and $U$, assuming there is minimal correlation among the variables. This is especially useful when using nonlinear embeddings such as delay embeddings in the regression. This changes the algorithm of DMDc but not significantly. In particular, we can now write:

$$\Omega = \begin{pmatrix} U_x & 0 \\ 0 & U_u \end{pmatrix} \begin{pmatrix} \Sigma_x & 0 \\ 0 & \Sigma_u \end{pmatrix} \begin{pmatrix} V_x^T \\ V_u^T \end{pmatrix} \tag{27}$$

This enables us to pick ranks separately for $X$ and $U$ components. In practice, we apply the techniques used in HAVOK (Brunton et al., 2017) to estimate the DMD. We do regression in the eigen-time-delay (pca-whitened) spaces of X and U (Hankelized), which allows us to select ranks separately for the X and the U space.

### B.1  ON NONLINEAR EMBEDDINGS IN DMDC

In the standard DMDc formulation (above), an SVD is taken across $\Omega$, which concatenates the state data $X$ with the input data $U$. Although this has the benefit of whitening across all regressors, it can bias the estimation of $A$ and $B$ depending on the relative scalings and dimensionalities of $X$ and $U$. This has a critical effect when applying high-dimensional nonlinear embeddings to only $X$ ($U$) individually, as the SVD will be increasingly dominated by signal from $X$ ($U$) if the data is sufficiently rich. Therefore, whenever we apply delay embeddings or other nonlinear embeddings to $X$, we do so commensurately to $U$.

### C  RELATIONSHIP BETWEEN DMD (REGULAR) AND DMDC

$$A_x^c = \left[ \begin{pmatrix} X^T X & X^T U \\ U^T X & U^T U \end{pmatrix}^{-1} \begin{pmatrix} X^T \\ U^T \end{pmatrix} \right]_{1:m} X_{n+1}$$

$$A_x = (X^T X)^{-1} X^T X_{n+1}$$

$$S = \left( U^T U - U^T X (X^T X)^{-1} X^T U \right)$$

$$A_x^c = \left[ \left( (X^T X)^{-1} + \overbrace{(X^T X)^{-1} X^T U}^{A_u} S^{-1} U^T X (X^T X)^{-1}, -(X^T X)^{-1} X^T U S^{-1} \right) \begin{pmatrix} X^T \\ U^T \end{pmatrix} \right] X_{n+1}$$

$$A_x^c = (X^T X)^{-1} X^T X_{n+1} + A_u S^{-1} A_u^T X^T X_{n+1} - A_u S^{-1} U^T X_{n+1}$$

$$A_x^c = A_x + A_u S^{-1} A_u^T X^T X_{n+1} - A_u S^{-1} U^T X_{n+1}$$

### D  PARTIAL OBSERVATION INDUCES BIASES IN INPUT OPERATOR B

Consider a partially observed linear system:

$$\begin{pmatrix} x^o \\ x^u \end{pmatrix}_t = \begin{pmatrix} A_{oo} & A_{ou} \\ A_{uo} & A_{uu} \end{pmatrix} \begin{pmatrix} x^o \\ x^u \end{pmatrix}_{t-1} + \begin{pmatrix} B_o \\ B_u \end{pmatrix} u_{t-1} \tag{28}$$

We observe states $x^o$. This system can also be formulated as a Vector-Autoregressive model with exogenous inputs (VAR-X). To see this formulation, we recursively substitute the definition of $x_t^u$ with its dynamical equation, hence arriving at a formulation of $x_t^o$ as a function of past observed states and inputs:

$$x_t^o = A_o x_{t-1}^o + A_{ou} x_t^u + B_o u_t \tag{29}$$

$$= A_o x_{t-1}^o + A_{ou}[A_{uo} x_{t-2}^o + A_{uu} x_{t-2}^u + B_u u_{t-1}] \tag{30}$$

$$= \ldots \tag{31}$$

$$= A_o x_{t-1}^o + \sum_{i=1}^{\infty} A_{ou} A_u^{i-1} (A_{uo} x_{t-1}^o + B_u u_{t-i}) + B_o u_t \tag{32}$$

We take an infinite sum here for completeness, but in practice $i$ can be capped up to marginal error based on the decay rates (eigenvalues) of $A_u$. We can write this equation as a function of the delay-embedded observed state and inputs:

$$x_{t+1}^o = \begin{bmatrix} A_o & A_{ou} A_{uo} & A_{ou} A_u A_{uo} & \ldots & A_{ou} A_u^{d-1} A_{uo} \end{bmatrix} \begin{bmatrix} x_t^o \\ x_{t-1}^o \\ \ldots \\ x_{t-d}^o \end{bmatrix} \tag{33}$$

$$+ \begin{bmatrix} B_o & A_{ou} B_u & A_{ou} A_u B_u \ldots A_{ou} A_u^{d-1} B_u \end{bmatrix} \begin{bmatrix} u_t \\ u_{t-1} \\ \ldots \\ u_{t-d} \end{bmatrix} \tag{34}$$

These equations show that when performing regression as in $DMDc$ on partially-observed, delay-embedded data, the estimates of $B$ become biased by the intrinsic dynamics in the unobserved states. Biases in $B$ emerges when utilizing delay embeddings as dimensionality expansions, as we can see from the above formulation. Although we display the formal connection with linear systems above, it is simple to observe that the same problem occurs with nonlinear dynamics as well.

# E DE-BIASING $B$ UNDER PARTIAL OBSERVATION WITH SUBSPACE IDENTIFICATION

In this section, we introduce SubspaceDMDc, a natural extension of two DMD models in the literature: Subspace DMD (Takeishi et al., 2017b) and DMDc (Proctor et al., 2016a). SubspaceDMDc has a notable difference from SubspaceDMD, as Takeishi et al. (2017b) utilize the subspace identification approach to handle observation noise, whereas we utilize subspace identification to handle input affecting future timesteps (although we gain noise robustness through similar means). In the control theory literature, there are a number of subspace identification algorithms, two of the most famous are Multivariable Output-Error State sPace (MOESP) modeling and Numerical Algorithms for Subspace State Space System Identification (N4SID) (Verhaegen & Verdult, 2007; Verhaegen, 1994; Van Overschee & De Moor, 1994). In order to be brief, we will discuss only N4SID, which is the method we chose to implement. In general, the algorithms have similar behavior, except on ill-conditioned data. Practically speaking, either method could be used in DSA; it is up to the user and their respective performances on the dataset. The extension of these methods to SubspaceDMDc is the introduction of a lifting feature space: polynomials, kernels, random feature maps, neural networks, or nonlinear features can be used in order to find a best-predicting nonlinear basis upon which the features evolve linearly.

## E.1 SUBSPACE DMD

Subspace DMD (Takeishi et al., 2017b) is designed to handle the estimation of the Koopman operator given data that is contaminated with observational and process noise. Assuming that the dynamics and the noise are independent, one can project out the contribution of the noise in the data and leave only the component that is explainable with past data (via delay embedding, step 2 of the algorithm below). We assume real data, although the method works for complex data as well. Algorithm 2 in the paper reads:

1. Construct data matrices $Y_p = \begin{bmatrix} Y_0^T & Y_1^T \end{bmatrix}^T$ $Y_f = \begin{bmatrix} Y_2^T & Y_3^T \end{bmatrix}^T$ where $Y_t = [g(x_t) \ldots g(x_{t-m+1})]$

2. Compute the orthogonal projection of the future data onto the past data: $O = Y_f \mathbb{P}_{Y_p^T}$ where the projector $\mathbb{P}_{Y_p^T} = Y_p^T (Y_p Y_p^T)^\dagger Y_p$.

3. Compute the compact SVD (e.g., the SVD with no zero rows or columns): $O = U_q S_q V_q^T$ and define $U_{q1}, U_{q2}$ by taking the first and last $n$ rows of $U_q$. This is done in order to split the projection matrix into the observability matrix and the state matrix: $O = \Gamma X$, up to right / left multiplication by an invertible matrix. The observability matrix looks like $\Gamma = (C \quad CA \quad \ldots \quad CA^n)$. Because this matrix encodes the time-shifted structure of the dynamics, we split into the top n and last n rows to get $U_{q1}$ and $U_{q2}$ upon which we do reduced-rank regression in the next step.

4. Compute the compact SVD of $U_{q1} = USV^T$ and define the operator $\tilde{A} = U^T U_{q2} V S^{-1}$.

5. If desirable, dynamic modes are defined as $w = \lambda^{-1} U_{q2} V S^{-1} \tilde{w}$ for eigenvalues $\lambda$, eigenvectors $\tilde{w}$ of $\tilde{A}$.

## E.2 N4SID

Numerical Algorithms for Subspace State Space System Identification (N4SID) (Van Overschee & De Moor, 1994) utilizes a similar approach as the above to jointly estimate $A, B, C, D$ operators in a state space model from data $Y$ and $U$. Here we briefly describe the algorithm that we apply to estimate $A$ and $B$ that are used for comparison of partially observed systems, as first defined by Overschee & Moor (1994). We used code from https://github.com/spmvg/nfoursid/tree/master for our implementation of n4sid. For the Subspace DMDc, we lift to a nonlinear space *before* state estimation.

For state estimation to succeed, standard conditions on the data state and input apply. In particular (1) the state vector is sufficiently excited (it explores all relevant dimensions of the state space), or the system is reachable, (2) the input sequence is persistently exciting, i.e., the Hankel matrix of the inputs is full rank, and (3) there is no linear state feedback, i.e. the state and the input are not collinear. Note that nonlinear feedback is permissible provided they are not collinear. Prediction in the SubSpaceDMDc is done with Kalman filtering, because state estimation must first take place.

Briefly, we explain the key computations behind N4SID. There are two slightly different approaches. The first algorithm is similar in spirit to Subspace DMD which we detail here:

### E.2.1 PROJECTION-BASED N4SID

As above, we create a Hankel data matrix of the observations, but also the input too, splitting this into past and future. First, we project out the data explained by $U_f$ in the future observations $Y_f$, but also the past observations and inputs $Z_p = [U_p \quad Y_p]$, thereby removing its influence. Then to remove measurement and process noise biases, we project the future states onto the space explainable by the states and inputs in the past, $Z_p$. This yields our matrix $O = \Gamma X$, which we split using SVD as before to get $\Gamma$, the extended observability matrix, and the states $X$ up to similarity. Noting again that our extended observability matrix has time-shifted structure, we can perform regression on the shifted components of $X$ given the instantaneous $U$, to arrive at $A, B$. The observability matrix $\Gamma$ also encodes $C$ in its top rows, which we can directly read out. However, we found this algorithm in practice to be less stable than the next one.

In pseudocode form, we have the following:

---

**Algorithm 1** Subspace DMD with Control (N4SID on lifted states)

---

**Require:** Output data $\mathbf{Y} \in \mathbb{R}^{p_{out} \times N}$, Input data $\mathbf{U} \in \mathbb{R}^{m \times N}$, past window $p$, future window $f$, system order $n$, regularization $\lambda$

**Ensure:** Estimated system matrices $\hat{\mathbf{A}}, \hat{\mathbf{B}}, \hat{\mathbf{C}}$

1: **procedure** BUILDHANKELMATRICES($\mathbf{Y}, \mathbf{U}, p, f$)
2:     $T \leftarrow N - p - f + 1$
3:     Construct Hankel matrices $\mathbf{Y}_p, \mathbf{U}_p, \mathbf{Y}_f, \mathbf{U}_f$
4:     $\mathbf{Z}_p \leftarrow \begin{bmatrix} \mathbf{U}_p \\ \mathbf{Y}_p \end{bmatrix}$
5:     **return** $(\mathbf{Y}_f, \mathbf{U}_f, \mathbf{Z}_p, T)$
6: **end procedure**

7: **procedure** OBLIQUEPROJECTION($\mathbf{Y}_f, \mathbf{U}_f, \mathbf{Z}_p, \lambda, T$)
8:     $\mathbf{\Pi}^{\perp}_{\mathbf{U}_f^T} \leftarrow \mathbf{I}_T - \mathbf{U}_f^T(\mathbf{U}_f\mathbf{U}_f^T + \lambda\mathbf{I})^{-1}\mathbf{U}_f$
9:     $\mathbf{Y}_{f,\perp} \leftarrow \mathbf{Y}_f\mathbf{\Pi}^{\perp}_{\mathbf{U}_f^T}$
10:     $\mathbf{Z}_{p,\perp} \leftarrow \mathbf{Z}_p\mathbf{\Pi}^{\perp}_{\mathbf{U}_f^T}$
11:     $\mathbf{O} \leftarrow \mathbf{Y}_{f,\perp}\mathbf{Z}_{p,\perp}^{\dagger}$                               ▷ Oblique projection via pseudoinverse
12:     **return** $\mathbf{O}$
13: **end procedure**

14: **procedure** ESTIMATESTATEFROMPROJECTION($\mathbf{O}, n$)
15:     $\mathbf{U}_o, \mathbf{S}_o, \mathbf{V}_o \leftarrow \text{SVD}(\mathbf{O})$
16:     Truncate to rank $n$: $\mathbf{U}_n, \mathbf{S}_n, \mathbf{V}_n$
17:     $\hat{\mathbf{\Gamma}}_f \leftarrow \mathbf{U}_n\sqrt{\mathbf{S}_n}$                               ▷ Estimated observability matrix
18:     $\hat{\mathbf{X}} \leftarrow \sqrt{\mathbf{S}_n}\mathbf{V}_n^T$                                 ▷ Estimated state sequence
19:     **return** $(\hat{\mathbf{\Gamma}}_f, \hat{\mathbf{X}})$
20: **end procedure**

21: $\mathbf{Y}_f, \mathbf{U}_f, \mathbf{Z}_p, T \leftarrow$ BUILDHANKELMATRICES($\mathbf{Y}, \mathbf{U}, p, f$)
22: $\mathbf{O} \leftarrow$ OBLIQUEPROJECTION($\mathbf{Y}_f, \mathbf{U}_f, \mathbf{Z}_p, \lambda, T$)
23: $\hat{\mathbf{\Gamma}}_f, \hat{\mathbf{X}} \leftarrow$ ESTIMATESTATEFROMPROJECTION($\mathbf{O}, n$)
24:
25:                                                         ▷ Align data for regression
26: $\hat{\mathbf{X}}_{\text{current}} \leftarrow \hat{\mathbf{X}}[:, 0:T-1]$
27: $\hat{\mathbf{X}}_{\text{next}} \leftarrow \hat{\mathbf{X}}[:, 1:T]$
28: $\mathbf{U}_{\text{mid}} \leftarrow \mathbf{U}[:, p:p+T-1]$
29:
30:                                                          ▷ Solve for system matrices
31: $\begin{bmatrix} \hat{\mathbf{A}} & \hat{\mathbf{B}} \end{bmatrix} \leftarrow \hat{\mathbf{X}}_{\text{next}} \begin{bmatrix} \hat{\mathbf{X}}_{\text{current}} \\ \mathbf{U}_{\text{mid}} \end{bmatrix}^{\dagger}$
32: $\hat{\mathbf{C}} \leftarrow$ first $p_{out}$ rows of $\hat{\mathbf{\Gamma}}_f$
33:
34: **return** $\hat{\mathbf{A}}, \hat{\mathbf{B}}, \hat{\mathbf{C}}$

---

# F MISALIGNED INPUT SPACES

For any orthogonal matrix $C$, the following equivalence holds:

$$y = Cx \iff \dot{y} = CA_1C^T x + CB_1 u(t) \tag{35}$$

Now consider the case where inputs are not equivalent in each system, but that they are also related by a coordinate transform:

$$u_y(t) = C_u u_x(t)$$

Then Eq. 38 resolves to:

$$\dot{y} = CA_1C^T x + CB_1C_u u_y(t) \tag{36}$$

This motivates the dissimilarity metric that seeks to jointly optimize $C$ and $C_u$, with the second term in Eq. 8 generalizing to

$$\min_{C_u \in O(n)} ||CB_1 C_u - B_2||$$

when $\alpha = 0$, equation 8 is the so-called two-sided Procrustes problem, which when solved jointly resolves to comparing the singular values of $B_1, B_2$: $||\Sigma_1 - \Sigma_2||$, which can be computed efficiently. When $\alpha \neq 0$, the two minimizations need to be jointly optimized. The method of optimization from Ostrow et al. (2023) can be effectively generalized to do so, with note that this is a larger optimization problem and requires longer optimization time (but see next section).

If the inputs that are directly applied to the system are known, as in RNN or RL models ($\dot{x} = f(x, u)$), this joint optimization procedure can be discarded. Likewise, when the inputs are aligned in time, Procrustes or other spatial alignment methods can be directly applied to the inputs first. Note that this input comparison does not directly compare the dynamics of the input, but rather how the input is read into the system. If one is interested in comparing the dynamics of the input as well, then DSA can be run on the input directly.

# G  SOLVING FOR OPTIMAL ORTHOGONAL $C$ EFFICIENTLY

The InputDSA formulation allows for efficient solving of the optimal $C \in O(n)$. Recall that

$$DSA(A_x, A_y) = \min_{C \in O(n)} \left|\left| A_x - CA_y C^T \right|\right|_F^2 \tag{37}$$

Is a non-convex optimization problem, and hence has to be solved iteratively Ostrow et al. (2023). However, the addition of the control constraint, $||B_x - CB_y||_F^2$ means that we can solve this problem using convex optimization for $\alpha = 0.5$. Observe that under similarity,

$$\tilde{A} = CAC^T, \tilde{B} = CB \implies \tilde{A}\tilde{B} = CAB$$

This suggests that we can identify $C$ via Procrustes alignment on the controllability matrix $K = \begin{pmatrix} B & AB & A^2 B & \dots A^n B \end{pmatrix}$, where $A \in \mathbb{R}^{n \times n}$:

$$\min_{C \in O(n)} ||K_1 - CK_2||_F^2 \tag{38}$$

The minimizer $C^*$ has a closed-form solution via orthogonal Procrustes. Likewise, jointly aligning the input dimension via $C_u$ (Appendix Sec. F) can be done in closed form as well via the two-sided Procrustes solution. This results in an acceleration of multiple orders, with the computation of $C$ taking O(1 millisecond), as opposed to O(1 second).

However, this formulation can result in $C^*$ that are biased towards the more controllable directions, i.e. $B$ can have an inordinate effect or can dominate. In practice, we found that using this approach with a ground truth $C$ resulted in the state similarity score becoming biased near dimension 30 (that is, $A \in \mathbb{R}^{30 \times 30}$). While this is still quite large, and the biases are small (average deviation $O(0.01)$ per element), we can do better. We can add further constraints to $C$, by noticing that $A^T$ also holds in the previous implication under similarity:

$$\tilde{A} = CAC^T, \tilde{B} = CB \implies \tilde{A}^T \tilde{B} = CA^T B$$

Thus, we can concatenate these powers as well to K, giving:

$$K = \begin{pmatrix} B & AB & A^T B & \dots & A^n B & A^{T^n} B \end{pmatrix}$$

Where the metric is once again Eq. 38. This improves the optimization stability on $A$ until at least dimension 150 for O(0.001) error per element error, which is more than enough in practice for InputDSA . We have the following lemma which states that this metric captures equivalency between two linear systems.

**Lemma G.1.** *Given two linear systems $x_{t+1} = A_x x_t + B_x u_t$ and $y_{t+1} = A_y y_t + B_y u_t$, Eq. 38 is equal to zero **if and only if** $y = Cx$ for some $C^T C = I$.*

*Proof.* Let us first consider the forward direction. Assuming $y = C^* x$, then we have the equivalence relationships $A_x = C^{*^T} A_y C^*$ and $B_x = C^{*^T} B_y$. Applying this relationship to $K_x$, we have

$$K_x = \left( C^{*^T} B_y \quad C^{*^T} A_y B_y \quad C^{*^T} A_y^T B_y \quad \ldots \quad C^{*^T} A_y^n B_y \right)$$
$$= C^{*^T} K_y$$

For which $\min_{C \in O(n)} ||K_x - CK_y||_F^2 = 0$ evidently at $C^*$.

Now consider the reverse direction. We can expand Eq. 38 as:

$$||K_x - CK_y||_F^2 = ||B_x - CB_y||_F^2 + ||A_x B_x - CA_y B_y||_F^2 + ||A_x^T B_x - CA_y^T B_y||_F^2 + \ldots$$

For $\min_{C \in O(n)} ||K_x - CK_y||_F^2 = 0$, each subterm must be zero for minimizer $\tilde{C}$. This immediately gives $B_x = \tilde{C} B_y$. Inspecting the next term, we substitute this relationship, giving

$$0 = ||A_x B_x - \tilde{C} A_y B_y||_F^2 = ||A_x \tilde{C} B_y - \tilde{C} A_y B_y||_F^2 \tag{39}$$
$$= ||(A_x \tilde{C} - \tilde{C} A_y) B_y||_F^2 \tag{40}$$
$$\implies A_x \tilde{C} = \tilde{C} A_y \tag{41}$$
$$\tag{42}$$

With the last step following from $B_y \neq 0$. This in turn gives $A_x = \tilde{C} A_y \tilde{C}^T$. We can similarly apply this reasoning to the next expression, which gives the same result. Reversing the previous logic, we have $x_{t+1} = \tilde{C} A_y \tilde{C}^T x_t + \tilde{C} B_y u_t \implies y = \tilde{C}^T x$. $\square$

For a given $A, B$, the standard right Procrustes problem is written as:

$$C^* = \text{argmin}_{C \in O(n)} ||CA - B||_F^2 \tag{43}$$
$$= \text{argmax}_{C \in O(n)} < CA, B >_F = \text{Tr}[(CA)^T B] \tag{44}$$

Writing the form of this problem with $K_1, K_2$, we can separate out individual elements in the Frobenius inner product, giving

$$< CK_1, K_2 >_F = \sum_{i=0}^{n} < CA_x^i B_x, A_y^i B_y >_F + < CA_x^{T^i} B_x, A_y^{T^i} B_y >_F \tag{45}$$
$$= \sum_{i=0}^{n} < C, A_y^i B_y (A_x^i B_x)^T >_F + < C, A_y^{T^i} B_y (A_x^{T^i} B_x)^T >_F \tag{46}$$
$$= < C, \sum_{i=0}^{n} A_y^i B_y (A_x^i B_x)^T + A_y^{T^i} B_y (A_x^{T^i} B_x)^T >_F \tag{47}$$

With the last steps due to linearity of the inner product and the second step using the trace permutation identity. This gives the maximum over $C \in O(n)$ to be

$$C^* = UV^T \quad \text{where} \quad \sum_{i=0}^{n} A_y^i B_y (A_x^i B_x)^T + A_y^{T^i} B_y (A_x^{T^i} B_x)^T = USV^T$$

In practice, taking large matrices $A$ to many powers results in numerical instability issues, especially when $\lambda_{max}(A) > 1$. Algorithmically, we check the condition number of $A^n$ before choosing to include the term in the controllability matrix. If it is too small or too large, we stop.

**Lemma G.2.** *In linear systems, controllability is only preserved under orthogonal changes of basis,* $y = Cx$, *where* $C \in O(n)$ *(if and only if).*

*Proof.* We define the controllability Gramian for the system $x_{t+1} = Ax_t + Bu_t$ as:

$$W_x = \sum_{i=0}^{T} A^i B (A^i B)^T = \sum_{i=0}^{T} A^i B B^T (A^i)^T \tag{48}$$

Consider the change of basis $y = Cx$. The dynamics become:

$$\begin{aligned} y_{t+1} &= C(Ax_t + Bu_t) \\ &= CA(C^{-1}y_t) + CBu_t \\ &= (CAC^{-1})y_t + (CB)u_t \end{aligned}$$

We compute the Gramian for $y$, denoted $W_y$, using the transformed matrices $(CAC^{-1})$ and $(CB)$. Note the identity $(CAC^{-1})^T = C^{-T}A^T C^T$.

Substituting these into the sum:

$$\begin{aligned} W_y &= \sum_{i=0}^{T} (CAC^{-1})^i (CB)(CB)^T ((CAC^{-1})^i)^T \\ &= \sum_{i=0}^{T} (CA^i \cancel{C^{-1}C} BB^T \cancel{C^T C^{-T}} (A^T)^i C^T) \\ &= C \left( \sum_{i=0}^{T} A^i BB^T (A^T)^i \right) C^T \end{aligned}$$

This simplifies to the congruence relation:

$$W_y = CW_x C^T \tag{49}$$

**Direction 1 (If):** If $C^T C = I$ (i.e., $C \in O(n)$), then $C^T = C^{-1}$. Equation (49) becomes:

$$W_y = CW_x C^{-1}$$

This implies $W_y \cong W_x$ (similarity), so the spectrum is preserved.

**Direction 2 (Only If):** If we require preservation of the spectrum, we must have $W_y \cong W_x$. By definition of similarity, this means there exists a relationship:

$$W_y = CW_x C^{-1} \tag{50}$$

Equating our derived result (49) with the requirement (50):

$$CW_x C^T = CW_x C^{-1}$$

Assuming $W_x$ is full rank (invertible), we can cancel $CW_x$ from the left:

$$C^T = C^{-1}$$

Multiplying by $C$ on the right:

$$C^T C = I$$

Thus, preservation requires $C$ to be orthogonal ($C \in O(n)$).

$\square$

## G.1 GENERALIZING THE WASSERSTEIN DISTANCE FOR INPUTDSA

Recall the Wasserstein distance over DMD eigenvalues,

$$\text{DSA}(\Lambda_1, \Lambda_2) := \min_{P \in \Pi(n)} ||P\Lambda_1 P^T - \Lambda_2||_F \tag{51}$$

This metric respects the notion of equivalency under general similarity transforms, $A \to CAC^{-1}$ for invertible C's, given that only eigenvalues are preserved under these transformations. We would like to identify a similar metric for input driven systems. To motivate our metric, consider applying a diagonalizing transform to the dynamics of our input-driven system:

$$x_{t+1} = Ax_t + Bu_t \tag{52}$$
$$= V\Lambda V^{-1} x_t + Bu_t \tag{53}$$
$$V^{-1} x_{t+1} = \Lambda V^{-1} x_t + V^{-1} Bu_t \tag{54}$$

We observe that the corresponding feature of the input to each eigenvalue $\lambda$ is the row vectors on $V^{-1}B$, which we henceforth term the eigenmode-input interaction matrix. We can easily show that these features are invariant to any invertible transform. Given a transform $A \to CAC^{-1}, B \to CB$,

$$V\Lambda V^{-1} \to CV\Lambda V^{-1}C^{-1} := \tilde{V}\Lambda\tilde{V}^{-1} \tag{55}$$

Hence $V \to CV$

$$V^{-1}B \to (CV)^{-1}CB = V^{-1}C^{-1}CB = V^{-1}B \tag{56}$$

Thus, a natural extension to Eq. 51 is the joint Wasserstein distance over $[\Lambda_i, (V^{-1}B)_i]$. Denoting $\Lambda^1$ the set of eigenvalues for system one, and denoting $\pi$ a permutation map,

$$\text{InputDSA}(\Lambda^1, \Lambda^2, V_1^{-1}B_1, V_2^{-1}B_2, \alpha) = \min_\pi \sum_i [\alpha(\Lambda_i^1 - \Lambda_{\pi(i)}^2)^2 + (1-\alpha)|(V_1^{-1}B_1)_i - (V_1^{-1}B_1)_{\pi(i)}|_2^2] \tag{57}$$

This metric is intuitive: $V^{-1}B$ describes how input direction interacts with the independent eigenmodes, which is related to the controllability of that mode. However, this metric has numerical stability issues. First, eigenvalues can only be identified up to an arbitrary phase-hence, we are forced to study instead the norms of each eigenmode-input interaction, $|(V_1^{-1}B_1)_i|_2$. This loses information but works reasonably for small systems. Identifying the eigenvectors of an arbitrary matrix is challenging for poorly-conditioned matrices. Hence, we suggest evaluating the conditioning of the DMD matrix before applying this metric.

## H INPUTDSA PSEUDOCODE

---
**Algorithm 2** InputDSA

---
**Require:** $X_1, X_2 \in \mathbb{R}^{n \times t \times d}$, $U_1, U_2 \in \mathbb{R}^{n \times t \times \ell}$, number of delays $q$, nonlinear lifting functions $\phi_1, \phi_2$
  1:      rank for state-space $r$,
**Ensure:** Similarity transform distance $d$ between the two dynamical systems

  2:  $A_1, B_1 \leftarrow \text{SUBSPACEDMDC}(\phi_1(X_1), \phi_2(U_1), r)$
  3:  $A_2, B_2 \leftarrow \text{SUBSPACEDMDC}(\phi_1(X_2), \phi_2(U_2), r)$
  4:  $d = \min_{\substack{C \in O(n) \\ C_u \in O(n)}} \alpha \|CA_1C^\top - A_2\|_F + (1-\alpha)\|CB_1C_u - B_2\|_F$
  5:  **return** $d$

---

## I HYPERPARAMETER TUNING FOR INPUTDSA

**Delay** In InputDSA , the delay parameter controls the size of the delay embedding used to estimate the dynamics operator. If too few delays are chosen, the embedding may distort the data and amplify noise. Conversely, too many delays fold the data into unnecessarily high dimensions, making it more difficult to model the dynamics with DMD (Ostrow et al., 2024).

---

**Algorithm 3** Dynamic Mode Decomposition with Control (DMDc)

---

**Require:** Delay-embedded states $H_X$, inputs $H_U$, truncation ranks $r_{\text{all}}, r_{\text{state}}$, ridge regularization $\lambda$
**Ensure:** Dynamics operator $A$ and $B$

1: **procedure** BUILDSNAPSHOTS($H_X, H_U$)
2:      $X_- \leftarrow H_X[:, 1:-1], \quad X_+ \leftarrow H_X[:, 2:]$
3:      $U_- \leftarrow H_U[:, 1:-1]$
4:      $\Omega \leftarrow \begin{bmatrix} X_- \\ U_- \end{bmatrix}$
5:      **return** $(X_+, X_-, U_-, \Omega)$
6: **end procedure**

7: **procedure** SVDS($X_+, \Omega$)
8:      $(U_p, \Sigma_p, V_p) \leftarrow \text{SVD}(\Omega)$
9:      Partition $U_p = \begin{bmatrix} U_{p1} \\ U_{p2} \end{bmatrix}$ into state/input blocks
10:      $(U_r, \Sigma_r, V_r) \leftarrow \text{SVD}(X_+)$
11:      **return** $(U_{p1}, U_{p2}, \Sigma_p, V_p, U_r)$
12: **end procedure**

13: **procedure** REDUCERANK($U_{p1}, U_{p2}, \Sigma_p, V_p, U_r$)
14:      Truncate to $r_{\text{all}}$: $U_{p1} \leftarrow U_{p1}[:, 1:r_{\text{all}}], U_{p2} \leftarrow U_{p2}[:, 1:r_{\text{all}}], V_p \leftarrow V_p[:, 1:r_{\text{all}}], \Sigma_p \leftarrow \Sigma_p[1:r_{\text{all}}]$
15:      Truncate to $r_{\text{state}}$: $U_r \leftarrow U_r[:, 1:r_{\text{state}}]$
16:      **return** $(U_{p1}, U_{p2}, \Sigma_p, V_p, U_r)$
17: **end procedure**

18: **procedure** COMPUTEOPERATORS($X_+, V_p, \Sigma_p, U_{p1}, U_{p2}, U_r, \lambda$)
19:      $\Sigma_p^\dagger(\lambda) \leftarrow \text{diag}\left(\frac{\sigma_i}{\sigma_i^2 + \lambda}\right)$
20:      $A \leftarrow X_+ V_p \Sigma_p^\dagger(\lambda) U_{p1}^\top$
21:      $B \leftarrow X_+ V_p \Sigma_p^\dagger(\lambda) U_{p2}^\top$
22:      Project to the state space $\tilde{A} \leftarrow U_r^\top A U_r, \quad \tilde{B} \leftarrow U_r^\top B$ **return** $(\tilde{A}, \tilde{B})$
23: **end procedure**

24: $X_+, X_-, U_-, \Omega \leftarrow$ BUILDSNAPSHOTS($H_X, H_U$)
25: $U_{p1}, U_{p2}, \Sigma_p, V_p, U_r \leftarrow$ SVDS($X_+, \Omega$)
26: $U_{p1}, U_{p2}, \Sigma_p, V_p, U_r \leftarrow$ REDUCERANK($\cdot$)
27: $A, B \leftarrow$ COMPUTEOPERATORS($X_+, V_p, \Sigma_p, U_{p1}, U_{p2}, U_r, \lambda$)
28: **return** $A, B$

---

**Rank** SubspaceDMDc involves one rank parameter $r$, corresponding to the dimensionality of the latent state space. In practice, selecting an $r$ slightly higher than the true state dimension often yields a better estimation of the $A$ matrix.

**Hyperparameter tuning pipeline** We suggest jointly optimize the delay and the rank $r$ according to the following criteria:

- **Prediction accuracy:** The delay embedding should enable accurate modeling of the dynamics. To evaluate this, we split the dataset into training and test sets, fit InputDSA (via SubspaceDMDc) on the training set, and assess performance on the test set using the mean absolute standardized error (MASE), a standard metric for time-series forecasting. MASE compares the forecast error of the model against that of a naïve persistence baseline predictor and is defined as

$$\text{MASE} = \frac{\frac{1}{T}\sum_{t=1}^{T}|y_t - \hat{y}_t|}{\frac{1}{T-1}\sum_{t=2}^{T}|y_t - y_{t-1}|}.$$

    A value MASE $< 1$ indicates that DMDc predicts next-step activity (using the estimated operators $A$ and $B$) more accurately than simply copying the current time step.

- **Model complexity:** The estimated operators $A$ and $B$ should not be overly complex or dominated by spurious features (e.g., many small eigenvalues clustered near zero). To assess this, we compute the Akaike Information Criterion (AIC) for next-step prediction on the test set. AIC balances predictive accuracy against model complexity and, in our setting, is given by

$$\text{AIC} = \ln\left(\frac{1}{N}\sum_{j=1}^{N}(x_j - y_j)^2\right) + \frac{2(r^2 + 1)}{N}.$$

Overall, we aim to select a rank that is small enough to avoid inflating the AIC, while still yielding good predictive accuracy (i.e., low MASE). In many low-dimensional dynamical systems, both AIC and MASE exhibit a characteristic elbow-shaped curve (for example, see Figure P.1 and Figure Q). We recommend selecting the rank at this elbow point, and then choosing a delay that yields a low MASE at that rank.

## J PARTIALLY OBSERVED SYSTEM COMPARISON FURTHER DETAIL

We discretely simulated the following equations (repeated from 14):

$$x_{t+1} = A(x_t + gF\tanh(x_t)) + B(u_t + \tanh(u_t)) \tag{58}$$

$$y_t = (\mathbf{I}_d \quad \mathbf{0}_{n-d})\, x_t + \epsilon_t \tag{59}$$

We generated two matrices $A_1, A_2$, sampling each element i.i.d. from a standard normal distribution. To enforce stability of these matrices, we globally rescaled the matrices by a term $\rho/\lambda_{max}$, where $\lambda_{max}$ is the max eigenvalue of the sampled matrix and $0 < \rho < 1$. We arbitrarily picked $\rho_1 = 0.92$ and $\rho_2 = 0.82$ to ensure a significant difference in the intrinsic dynamics, but not so large as to make the data obviously different. We set $g = 0.1$ for each system, and fixed $F$ to be the matrix defined as $F_{ij} = \delta_{ij}\delta_{i\leq d}$ where $d$ is the number of observed states in the observation matrix $C = (\mathbf{I}_d \quad \mathbf{0}_{n-d})$. We sampled $B_1, B_2$ from normal distributions as well, with $B_{1_{ij}} \sim N(0, g_1)$, $B_{2_{ij}} \sim N(0, g_2)$, setting $g_1 = 0.5, g_2 = 2.0$. We sampled $\epsilon_i \sim N(0, 0.01)$ for each observed index for each timepoint.

Across Figs. 2b,c,d,e, we simulated 20-dimensional systems with only 2 dimensions observed, for 5000 timepoints. For every type of DMD, we applied delay embeddings of size 150, and fit state space / dynamics matrices with rank 20. We chose these parameters by inspecting the spectral distribution of the estimated observability matrix (line 16 of Algorithm 1) across multiple delays. We added delays under the largest modes before the spectral drop-off point stopped changing (similar to the idea of a false neighbors analysis, Kennel et al. 1992), then picked the elbow of that curve. We selected the maximum of those values for each of the four systems. We observe these curves in Fig.

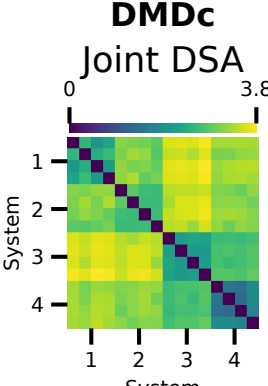

Figure 6: **Joint InputDSA Comparison using DMDc and SubspaceDMDc** The sum of jointly-optimized state and input distances is presented here, with $\alpha = 0.5$. Comparisons were generated on the same dataset as in Fig. 2c and d. DMDc Silhouette score on state, input: 0.235, 0.088. SubspaceDMDc Silhouette score on state, input: 0.368, 0.55.

7. However, we note that InputDSA is robust to a number of different ranks (Fig. 8), both larger and smaller than the true system size.

We computed silhouette score using Scikit-Learn on the precomputed InputDSA distance. Based on some given label (here, state or input ground-truth similarity), the dataset is divided into subsets $C_1, C_2, \ldots C_n$ with each data point $x_1, x_2, \ldots x_N$ belonging to one subset. Define the labels (cluster index) of each point as $c_1, c_2, \ldots c_N$. Next, the mean intra-cluster and the minimum mean inter-cluster distance is computed for each data point:

$$a(i) = \frac{1}{|C_{c(i)}|} \sum_{C_{c(j)} = C_{c(i)}, i \neq j}^{N} d(x_i, x_j)$$

$$b(i) = \min_{j \neq c(i)} \frac{1}{|C_j|} \sum_{c(k) = j}^{N} d(x_i, x_k)$$

Where $|C_{c(i)}|$ denotes the cardinality of the set, and $d(\cdot, \cdot)$ denotes the distance function to be used. In our setting, we use the InputDSA input or state distances for $d$. Lastly, the silhouette score is computing as:

$$S = \frac{1}{N} \sum_{i=1}^{N} \frac{b(i) - a(i)}{\max(a(i), b(i))}$$

The silhouette score approaches 1 when all points in each class are strongly separated and there is minimal distance between the points within each class, while it is 0 if the inter- and intra-cluster distances are equivalent. It is notable that a silhouette score of 0.7 can correspond to perfect linear classification of all classes, as deviations from 1.0 can be caused by within-class variance that remains non-overlapping with other classes.

## J.1 ROBUSTNESS TO HYPERPARAMETERS

Here, using partially observed nonlinear systems with known ground-truth similarity structure, we test InputDSA's robustness to changes in hyperparameters. For 20-dimensional systems with only 2 observed dimensions, we vary both the delay and the rank used in fitting InputDSA. As shown in Fig. 8, rank and delay affect state similarity more than input similarity, while the Silhouette

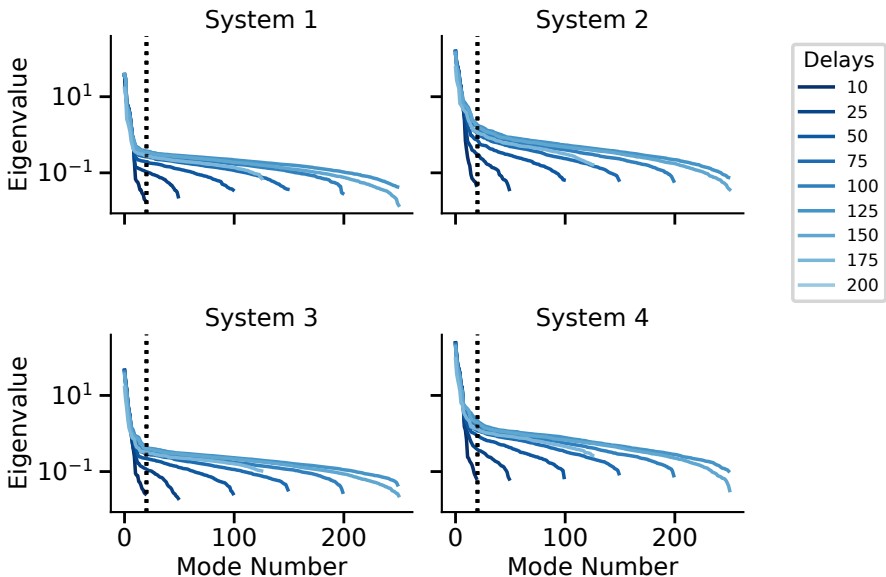

Figure 7: Spectral content of the Extended Observability Matrix from Subspace DMDc for each system in Fig. 2b across multiple delays. Dotted line indicates rank 20.

score for input similarity remains nearly invariant across hyperparameters. When the chosen rank is too small relative to the true system dimensionality ($N = 20$), the learned dynamics operator $A$ cannot adequately capture the state dynamics, yielding low Silhouette scores on the fitted distance matrix. Once the rank is large enough to capture the high-variance directions of the system, even if still lower than the true dimensionality, the Silhouette score approaches 1, indicating nearly perfect clustering accuracy. In other words, the rank used for InputDSA does not need to match the ground-truth dimensionality; it only needs to be high enough to capture the dominant dynamical modes. Beyond this threshold, the estimated similarity structure becomes robust to small increases in rank. However, in real-world noisy systems, choosing a rank that is too high may incorporate spurious, noise-driven dimensions into the dynamic operator. We therefore recommend selecting rank via a hyperparameter sweep as described in Appendix I.

For the delay, following (Ostrow et al., 2024), choosing too few delays can distort the embedding and amplify noise, whereas too many delays fold the data into unnecessarily high dimensions, making it harder for DMD to model the dynamics effectively. Here, we demonstrate that InputDSA's estimated similarity structure is robust to delay choice within a reasonable range—as long as the delay is sufficiently large for the delay-embedded state to span a rich enough basis to fit a linear dynamic operator.

### J.2 EFFECT OF INPUT DIMENSIONALITY ON SYSTEM IDENTIFICATION

In Fig. 2, we analyzed partially observable systems driven by low-pass filtered one-dimensional white noise. Here, we vary the dimensionality of the input and evaluate how the Silhouette score (computed against ground-truth labels) changes as partial observability increases, using InputDSA distance estimates based on Subspace DMDc. As shown in Fig. 9, increasing the dimensionality of the input provides only a modest improvement in recovering the latent state similarity structure, while simultaneously reducing the identifiability of the input-driven dynamics.

## K  PROCESS NOISE EFFECT ON INPUTDSA

In real-world systems such as neural circuits, process noise is pervasive: intrinsic stochasticity in the biophysics, fluctuations in synaptic and network activity, and unobserved perturbations all contribute to variability in the dynamics. We tested InputDSA's robust to process noise added to the state in

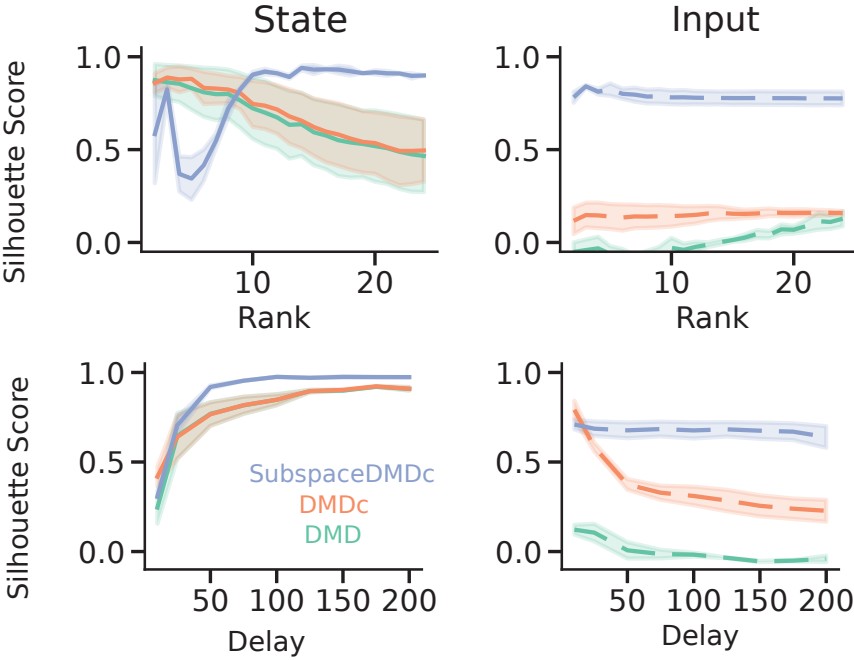

Figure 8: Effect of rank and delay in each DMD algorithm on clustering scores, utilizing 20 dimensional systems with 2 dimensions observed, and 1000 datapoints per dataset. When varying the rank, we fix the number of delays to be 150 and when varying the delay, we fix the rank to be the true rank of the system, i.e. $N = 20$.

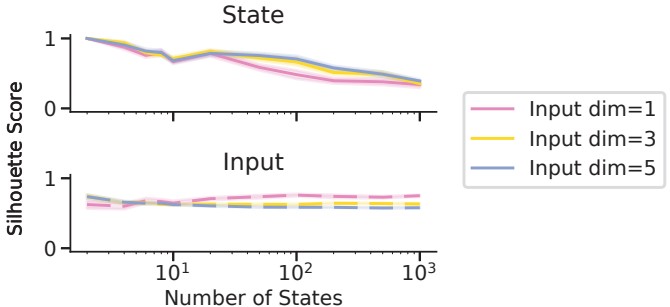

Figure 9: **Effect of input dimensionality on InputDSA performance under partial observability.** Silhouette scores for InputDSA based on Subspace DMDc as the system is increased from 2-dimensional to 1000-dimensional, while only 2 dimensions observed. Each size was repeated across 20 random seeds. Shading denotes standard error.

Section 3.1. We again consider the partially observed nonlinear dynamical system parameterized by this set of equations:

$$x_{t+1} = A(x_t + gF \tanh(x_t)) + B(u_t + \tanh(u_t)) + \eta_t \tag{60}$$

$$y_t = (\mathbf{I}_d \quad \mathbf{0}_{n-d}) \, x_t + \nu_t \tag{61}$$

Here, $\nu_t \sim \mathcal{N}(0, \sigma^2)$ is the process noise added to the latent state at time $t$. We generate the data exactly as in Section 3.1. The state-space similarity structure estimated by InputDSA remains highly robust to process noise, exhibiting an almost unchanged Silhouette score as $\sigma$ increases (Fig. 10B). On the other hand, the Silhouette score of the input-driven similarity structure decreases smoothly

with increasing $\sigma$, reflecting a graceful degradation in the identifiability of input-driven dynamics under higher process noise.

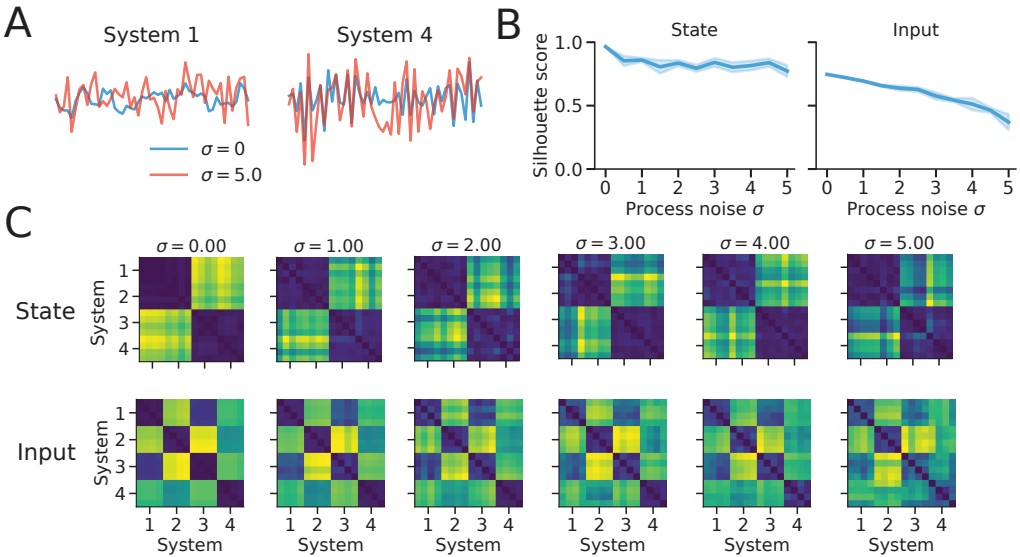

Figure 10: **InputDSA remains robust to moderate levels of process noise.** (A) Example latent state activity $x_t$ over time sampled from two different dynamical systems, overlaid with the state trajectory under process noise sampled from $\mathcal{N}(0, \sigma)$ where $\sigma = 5.0$. (B) Silhouette score against the groundtruth label when the systems are clustered based on the InputDSA state or input similarity sturcture. (C) Example distance matrices underlying varying levels of $\sigma$.

## L   ESTABLISHING A NOISE FLOOR FOR INPUTDSA SCORE

It is often desirable to establish a statistically insignificant noise floor for the InputDSA score, below which two systems can be considered effectively identical. To estimate this noise floor, we can split the state and input time series of a single system into two halves (or multiple segments, if sufficient data are available) and compute the InputDSA score between these segments. The resulting within-system scores provide an empirical noise floor against which cross-system comparisons can be evaluated.

Here, using the partially observed nonlinear system examples in Fig. 2, we computed and compared three quantities for the state distance and input distance (1) *Split-half*: the split-half noise floor, established by computing the distance between segments of state and input trajectory of the same system driven by identical input, (2) *Same:* the distance between systems sharing identical groundtruth state / input dynamic operator, but driven by inputs with different random initial conditions, (3) *Different*: the distance between systems with different state / input dynamic operator. In other words, both *Split-half* and *Same* are within-system distances, while *Different* measures cross-system distances estimated by differnt methods.

As shown in Fig. 11, the *Split-half* and *Same* distances are typically similar, and both are markedly smaller than the *Different* distances. This provides a basic sanity check that the comparison methods tested here can reliably distinguish between different dynamical systems. One exception is the input distance estimated by DMDc, where the *Split-half* and *Different* scores are comparable, indicating that DMDc fails to recover input-driven dynamics correctly in partially observed nonlinear systems.

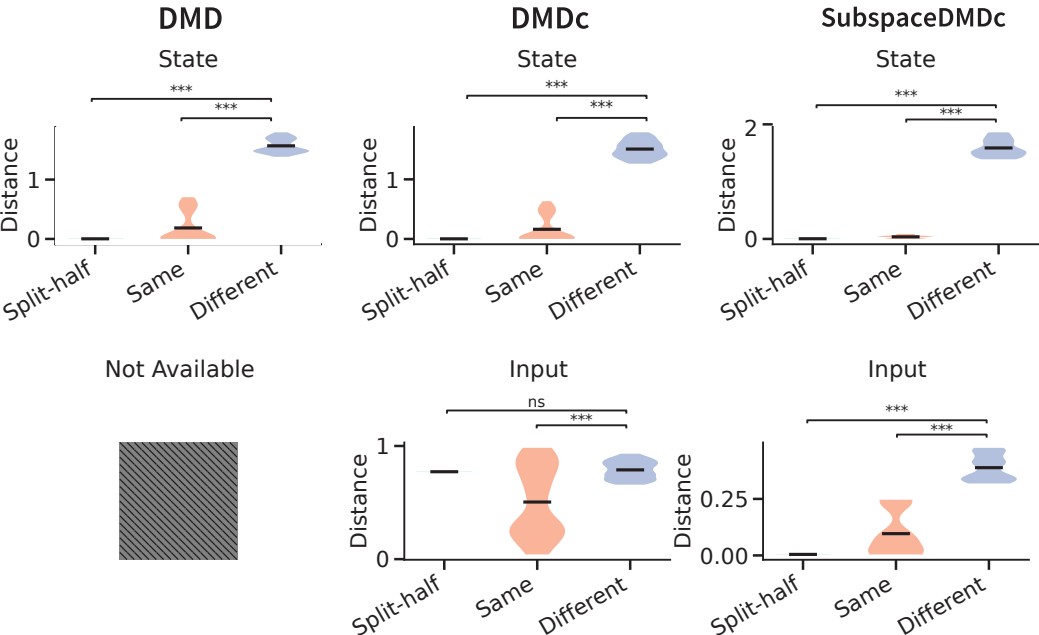

Figure 11: **Noise floor analysis for state and input distances in partially observed nonlinear systems**. We compare three quantities: (1) *Split-half*, the within-system noise floor estimated by comparing segments of the same trajectory; (2) *Same*, the distance between systems sharing identical state/input dynamics but driven by different input initializations; and (3) *Different*, the distance between systems with different underlying dynamics. For both state and input distances, *Split-half* and *Same* are similarly small and substantially below *Different*, demonstrating that the tested methods reliably distinguish different dynamical systems. DMDc is an exception for input distance, where *Split-half* and *Different* are comparable, indicating poor recovery of input-driven dynamics under partial observability.

## M    INPUT NOISE GENERATION

To assess the robustness of InputDSA to noisy or corrupted inputs, we systematically added different types of noise or transformations to the input time series of the nonlinear dynamical systems we created. Below we describe how each type of noise was generated. We visualize examples of the noise-corrupted input in Fig. 12 and Fig. 13 .

**Gaussian (white) noise.**    White Gaussian noise was added independently to each input channel:

$$\tilde{u}(t) = u(t) + \mathcal{N}(0, \sigma^2),$$

where $\sigma$ is set by the noise level.

**Pink noise.**    Pink $(1/f)$ noise was generated in the frequency domain with power spectrum proportional to $1/f^\alpha$ (with $\alpha = 1$ by default), then inverse Fourier transformed and scaled to the desired amplitude.

**Rotation.**    For two-dimensional input signals, we applied a random planar rotation:

$$\tilde{u}(t) = R(\theta) \, u(t), \quad R(\theta) = \begin{bmatrix} \cos\theta & -\sin\theta \\ \sin\theta & \cos\theta \end{bmatrix},$$

with rotation angle $\theta$ proportional to the noise level.

**Low-pass filtering.**    Inputs were smoothed using a digital Butterworth low-pass filter with cutoff frequency set by the noise level. Larger values corresponded to stronger filtering.

**Multiplicative noise.**    Each input channel was scaled by a random Gaussian factor:

$$\tilde{u}(t) = u(t) \cdot \eta, \quad \eta \sim \mathcal{N}(1, \sigma^2),$$

where $\sigma$ is set by the noise level.

**Uniform noise.**    Additive noise sampled uniformly from $[-a, a]$ was added to each channel, where $a$ is the noise level.

**Impulse noise.**    At each time point, with probability $p$, an impulse of magnitude $\pm\alpha$ (set by the noise level) was added to the input.

**Baseline drift.**    A slow oscillatory drift was added to each channel:

$$d(t) = A \sin(2\pi f t) + \tfrac{A}{2} \sin(4\pi f t),$$

where $A$ is the drift amplitude (noise level) and $f$ is a low drift rate.

**Partial observability.**    A random fraction of input time series was masked with zeros, with masking probability given by the noise level.

Table 1: Noise levels used in experiments for each noise type.

| Noise type | Levels used |
|---|---|
| White (Gaussian) | 0.1, 0.5, 1, 1.2, 1.5, 2, 3 |
| Pink | 0.01, 0.1, 0.2, 0.3, 0.5, 0.7, 0.9, 1 |
| Rotation | 0.2, 0.5, 0.8, 1 |
| Low-pass | 0.2, 0.5 |
| Multiplicative | 0.1, 0.2, 0.5, 1, 1.2, 1.5, 2, 3 |
| Uniform | 0.01, 0.1, 0.5, 1, 1.2, 1.5, 2, 3 |
| Impulse | 0.5, 1, 2, 3, 4 |
| Baseline drift | 0.5, 1, 1.2, 1.5, 2, 3, 4 |
| Partial observability | 0.1, 0.3, 0.5, 0.7, 0.9, 0.98 |
| Temporal smoothing | 5, 10, 20, 30, 40, 50 |

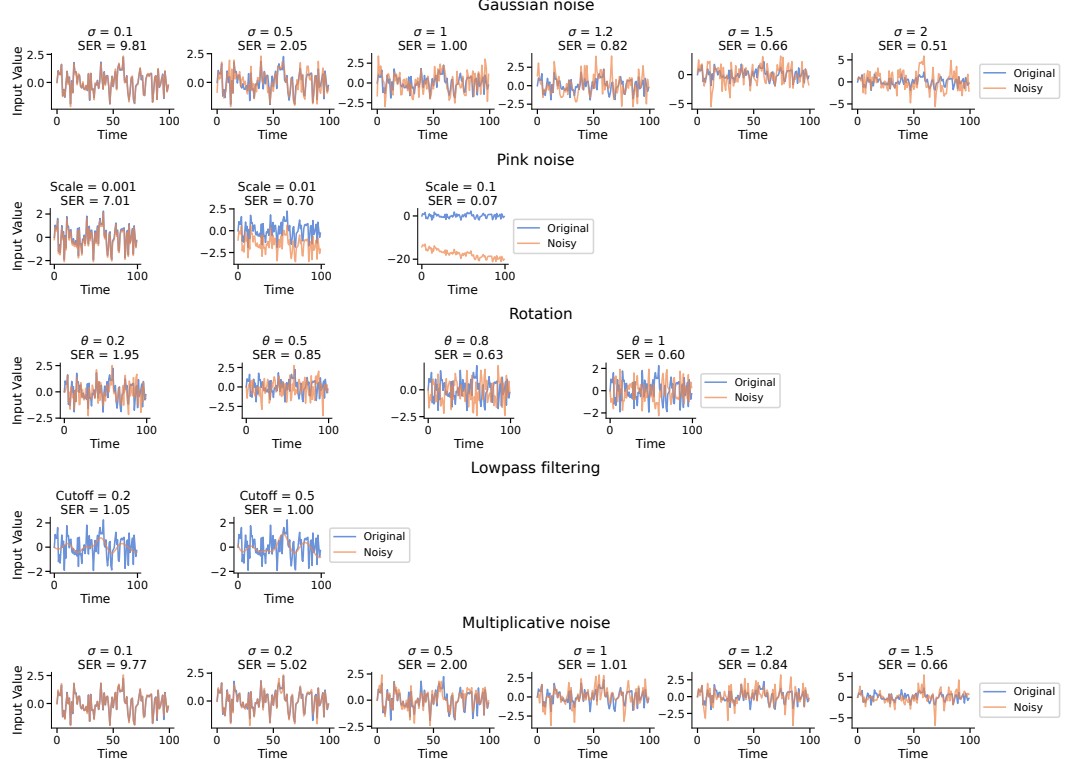

Figure 12: Effect of different noise types and levels on the input time series of the nonlinear dynamical system as in Fig. 2. The plots show activity along four example observation dimensions received by the networks during an example trial. Here, we show gaussian white noise, pink noise, rotation, low-pass filter, and multiplicate noise applied to the input.

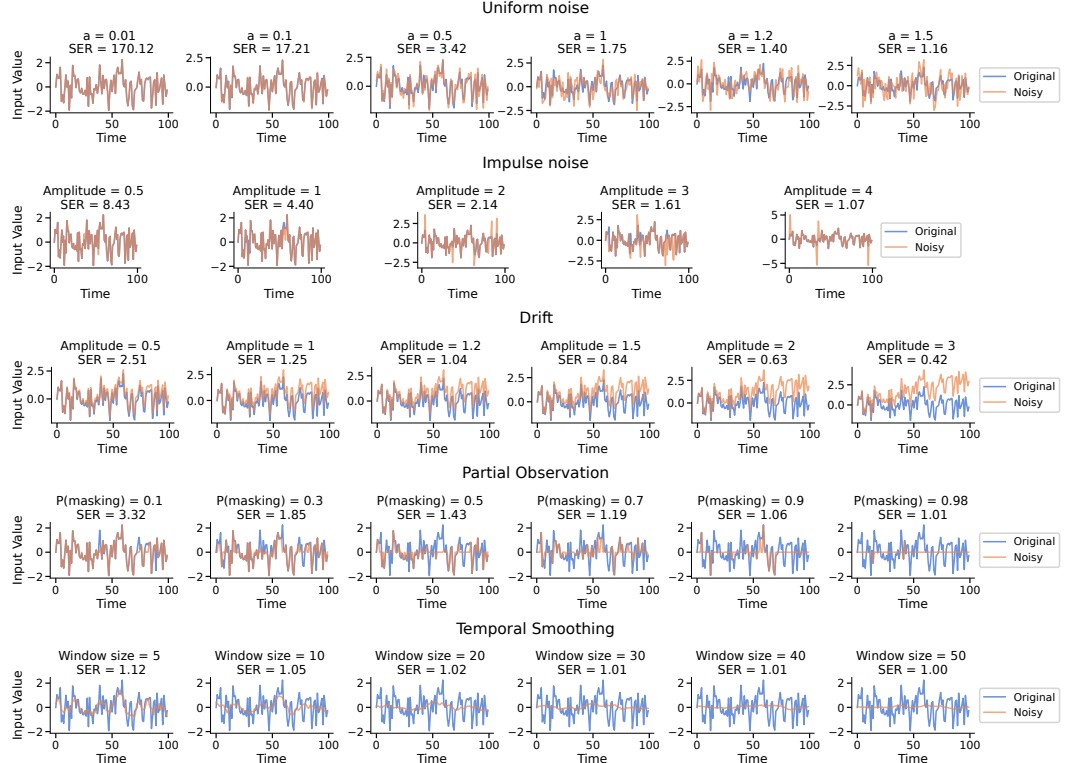

Figure 13: Effect of different noise types and levels on the input time series of the nonlinear dynamical system as in Fig. 2. The plots show activity along four example observation dimensions received by the networks during an example trial. Here, we show noise sampled from an uniform range, impulse noise, random drift, partial observability, and temporal smoothing applied to the input.

## M.1 EFFECT OF TIME-SHIFTED INPUT ON INPUTDSA DISTANCE ESTIMATION

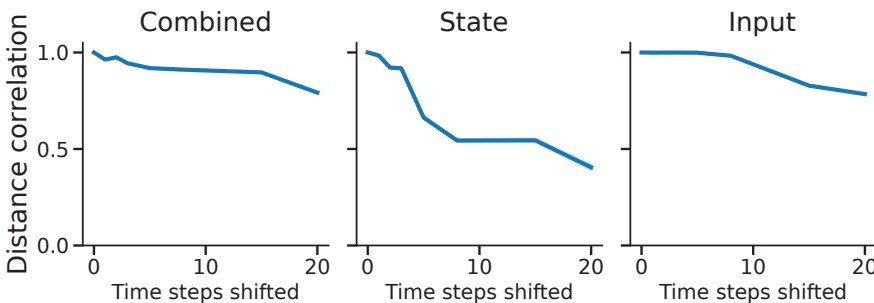

Figure 14: Correlation between the InputDSA distance matrix estimated using the ground truth input, and using time-shifted version of the true input.

# N   ORDINARY LEAST SQUARES BIASES ESTIMATES OF $A$ IN THE PRESENCE OF INPUT NOISE

Using two datasets: the nonlinear dynamical systems as in Fig. 2, and the RNNs trained on Random Target Task as in Fig. 3, we show that increasing the noise variance systematically contracts the singular spectrum of $B$ toward zero. Because the true input effect is underfit, the SubspaceDMDc regression inflates the real part of $A$'s eigenvalues to absorb the variance in the inputs that correlates with the state.

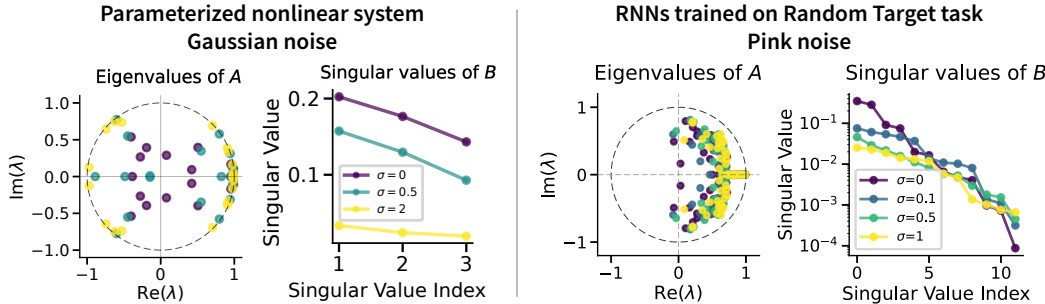

Figure 15: The eigenspectrum of the $A$ operator and singular spectrum of the $B$ operator when the input time-series is corrupted by Gaussian or pink noise of different variance.

Here, we also present a short theoretical discussion of this effect.

We consider the true system
$$Y = AX + BU,$$
where

- $Y \in \mathbb{R}^{n \times T}$ are the next states,
- $X \in \mathbb{R}^{n \times T}$ are the current states,
- $U \in \mathbb{R}^{m \times T}$ are the inputs,
- $A \in \mathbb{R}^{n \times n}$ and $B \in \mathbb{R}^{n \times m}$.

We observe noisy inputs
$$\tilde{U} = U + E,$$
where $E$ is input noise. The regression becomes
$$Y \approx \hat{A}X + \hat{B}\tilde{U}.$$

We solve the regression problem with OLS by first stacking the regressors:
$$Z = \begin{bmatrix} X \\ \tilde{U} \end{bmatrix},$$

The OLS estimator is
$$\begin{bmatrix} \hat{A} & \hat{B} \end{bmatrix} = YZ^\top (ZZ^\top)^{-1}.$$

Expanding $Z$,
$$ZZ^\top = \begin{bmatrix} X \\ \tilde{U} \end{bmatrix} \begin{bmatrix} X^\top & \tilde{U}^\top \end{bmatrix} = \begin{bmatrix} XX^\top & XU^\top + XE^\top \\ UX^\top + EX^\top & UU^\top + UE^\top + EU^\top + EE^\top \end{bmatrix}.$$

Assuming $E$ is zero-mean and independent,
$$\mathbb{E}[ZZ^\top] = \begin{bmatrix} \Sigma_{xx} & \Sigma_{xu} \\ \Sigma_{ux} & \Sigma_{uu} + \Sigma_{ee} \end{bmatrix},$$

$$\mathbb{E}[YZ^\top] = \begin{bmatrix} \Sigma_{xx}A^\top + \Sigma_{xu}B^\top \\ \Sigma_{ux}A^\top + \Sigma_{uu}B^\top \end{bmatrix}.$$

To gain intuition, we consider the scalar case where

$$\sigma_{xx} = \text{Var}(x), \quad \sigma_{uu} = \text{Var}(u), \quad \sigma_{xu} = \text{Cov}(x, u), \quad \sigma_{ee} = \text{Var}(E).$$

The least-squares estimates are

$$\hat{a} = a + b \cdot \frac{\sigma_{xu}\,\sigma_{ee}}{\sigma_{xx}(\sigma_{uu} + \sigma_{ee}) - \sigma_{xu}^2},$$

$$\hat{b} = b \cdot \frac{\sigma_{xx}\sigma_{uu} - \sigma_{xu}^2}{\sigma_{xx}(\sigma_{uu} + \sigma_{ee}) - \sigma_{xu}^2}.$$

We can see that for $\hat{b}$, large $\sigma_{ee}$ in the denominator attenuates $\hat{b}$ toward zero. For $\hat{a}$, the direction of the bias is dependent on the signs of $b$, $\sigma_{xu}$ and relative weights of $\sigma_{xx}$ and $\sigma_{xu}$. In particular, when $b > 0$ and $\sigma_{ee}$ is large, $\hat{a}$ can be inflated. Intuitively, when the state and input are strongly positively correlated and the input drives the state in the same direction, $\hat{a}$ can be overestimated to absorb the shared variance in the input.

## O    RANDOM TARGET REACH TASK

We trained recurrent neural network (RNN) policies to perform a random target reaching task in the MotorNet simulation environment (Codol et al., 2024b). We used code from https://github.com/motornet-org/MotorNet. The effector was a `ReluPointMass24` model, a 2D point-mass skeleton attached to 4 muscles and controlled by muscle activations. The environment provided a sequence of random goals and fingertip states. The objective of the policy was to minimize the distance between fingertip position and target over the course of each episode. At each time step, the model receives a 12-dimensional observation consisting of the proprioceptive input, visual input, and the last action taken by the model. The action space is 4-dimensional consisting of the activation of each muscle. Each network consisted of a single recurrent layer (64 hidden units) followed by a linear readout and sigmoid nonlinearity to produce bounded muscle activations. Training was carried out using the Adam optimizer with a learning rate of 0.001. The loss function was the mean L1 distance between fingertip trajectories and target trajectories across timesteps. We visualize the groundtruth input alongside different types of surrogate inputs during an example trial in Fig. 16.

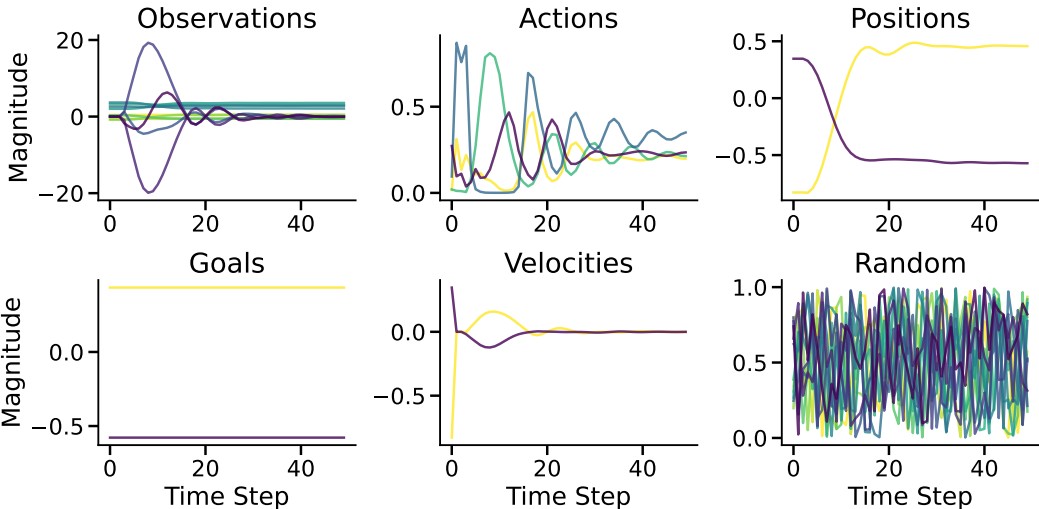

Figure 16: The true input (Observations) and different types of surrogate inputs during an example trial of the Random Target task.

## P    PLUME TRACKING TASK

We used the plume tracking task implemented in Singh et al. (2023) and the training code in https://github.com/BruntonUWBio/plumetracknets. In short, the plume tracking environment is a

2D arena where an odor source emits puffs carried downwind by a steady flow. The wind can be constant, switch once, or switch multiple times during a trial. Each puff diffuses and drifts, producing intermittent odor encounters like in real plumes. The agent uses an actor–critic architecture with a vanilla RNN backbone, followed by separate two-layer MLPs for the actor and critic. At each timestep it receives three inputs: egocentric wind direction along the x-axis, wind direction along the y-axis, and local odor concentration. Based on its internal state, the actor outputs a two-dimensional action specifying turn rate and forward speed.

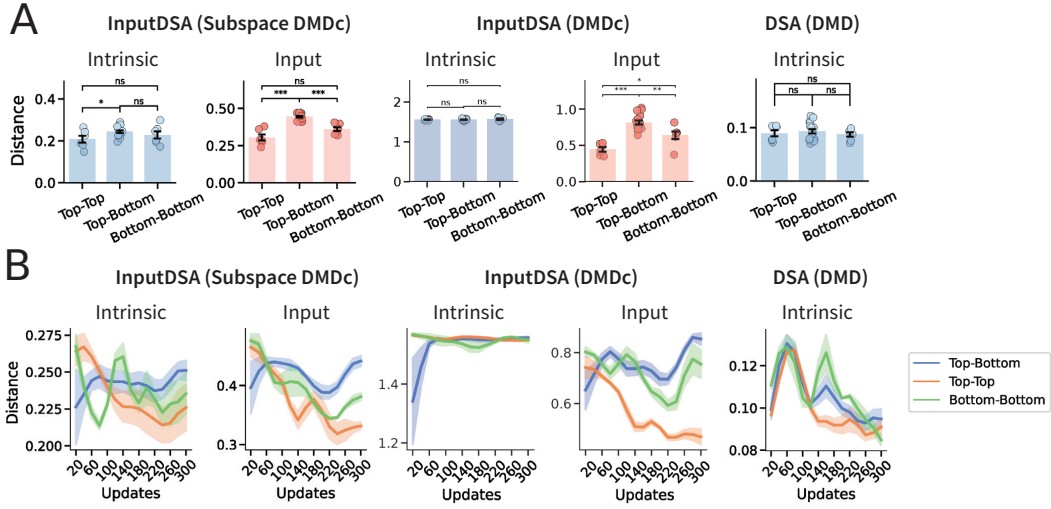

Figure 17: Full comparison of applying Subspace DMDc, DMDc, and DMD to the plume tracking dataset. **(A)** Average distance computed within the 5 Top Agents, within the 5 Bottom agents, and across groups (Top–Bottom). **(B)** The evolution of similarity within and across groups over learning. Shaded area indicates standard error.

### P.1 HYPERPARAMETER SWEEPING

To assess the optimal hyperparameters to use for InputDSA, we conducted a sweep of ranks and delays, as described by Sec. I. By picking the minimum / elbow of the prediction error curves (AIC, MASE), we choose a delay of 40 and rank of 50 for all InputDSA computations on this dataset. We also computed a non-normality score (the commutator score, $||AA^T - A^T A||_F^2$, which mea-

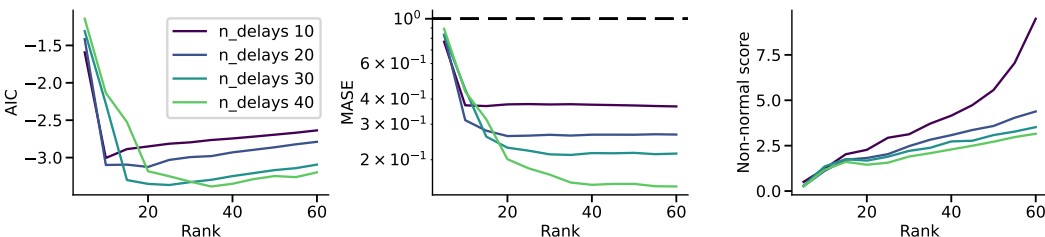

Figure 18: Hyperparameter sweep over number of delays and model rank on the plume tracking dataset.

sures (as described) the degree to which a matrix is non-normal (with normality being defined as $A^T A = AA^T$). This measures the relevancy of non-normality in the prediction of the SubspaceD-MDc model, which motivates the use of aligning the dynamics over orthogonal matrices, rather than invertible matrices – although any invertible matrix is a coordinate transform, the dynamical system's transient before can change when the transform is non-orthogonal. Hence, capturing the full dynamical similarity of two systems can entail comparing up to orthogonal transform in these settings. Here, we find that a rank of 50 with a delay of 40 has a non-normality score close to 2.5, indicating that transient dynamics can be significantly effected.

# Q NEURAL DATASET

The dataset published with Luo et al. (2025) can be found here: https://datadryad.org/dataset/doi:10.5061/dryad.sj3tx96dm.

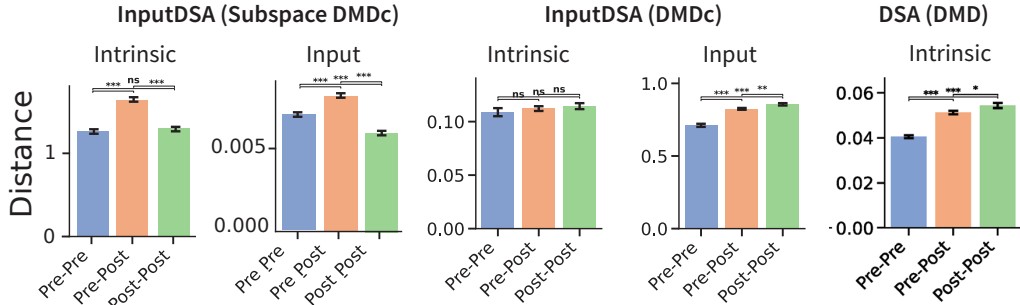

Figure 19: Full comparison of applying Subspace DMDc, DMDc, and DMD to the Luo et al. (2025) dataset. Similarity of neural dynamics before and after the nTc for rat T223. Bars denote standard error across 21 sessions.

## Q.1 HYPERPARAMETER SWEEPING

We chose a delay of 5 for the delay embedding and a rank of 6 for the reduced-rank regression on this dataset.

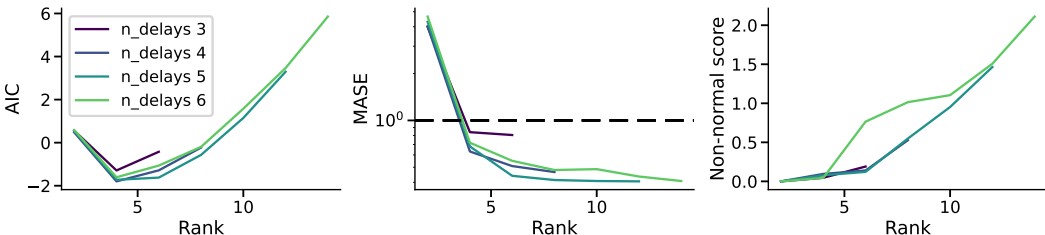

Figure 20: Hyperparameter sweep over number of delays and model rank on the processed Luo et al. (2025) dataset.

## Q.2 SURROGATE INPUTS USED IN NEURAL DATASET ARE PERSISTENTLY EXCITING

To assess whether these inputs were rich enough to robustly model the input operator, we computed the standard measure of persistent excitation used in control theory to answer this question. An input is persistently exciting if the matrix

$$M(t) = \int_0^t \tilde{u}(t)\tilde{u}(t)^T dt$$

is positive definite, i.e., the covariance matrix of the embedded input is full rank. In this analysis, we used models with 5 delays and rank 6 (neural states were 3 dimensional and inputs were 2 dimensional), so this covariance matrix needs to be at least rank 6. We computed this using the delay-embedded input as $\tilde{u}$ for each session of two rats, 39 in total with 100 trials on average per session (Fig. 21). In each session, the 6th singular value of this covariance matrix is much greater than 0 for each session, with the 10% quantile having value of 0.22 and the 1% quantile being 0.01. This suggests that even the input is simple, it is persistently exciting enough to robustly identify a useful B operator, given that there are a large number of trials.

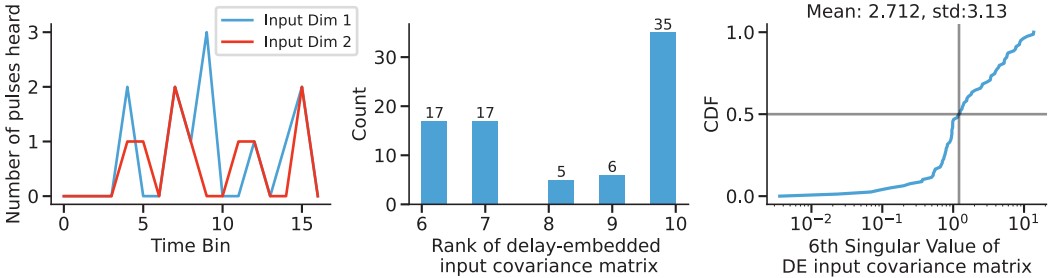

Figure 21: Persistent Excitation analysis of the Luo et al. (2025) surrogate input. The input data represents the total number of pulses coming from each speaker within each time bin of the trial. Left: sample input for one trial. Middle: distribution of the ranks of the embedded input covariance matrix for 80 individual sessions across 4 rats. SubspaceDMDc models are chosen to be rank 6, so all sessions are persistently exciting. Right: Cumulative distribution of the 6th singular value of the embedded input covariance matrix, indicating that these values are significantly greater than 0 for all sessions.

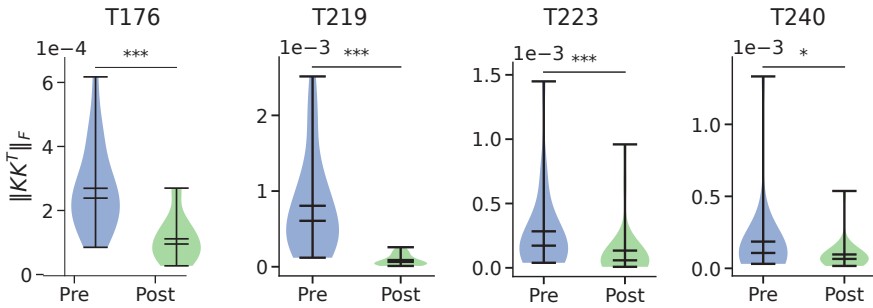

Figure 22: Distribution of Frobenius norms of Controllability Gramians for SubspaceDMDc models for all sessions of all rats. * represents $p < 0.05$ and *** represents $p < 0.001$ using Mann Whitney U-Test.

## Q.3    CONTROLLABILITY ENERGY FOR ALL RATS

## Q.4    SUBSPACE ANGLE

Given two dynamics matrices $A_x, A_y$, an orthonormal basis for each $(\tilde{A}_x, \tilde{A}_y)$ is first computed (for example, via SVD or QR decomposition). Then, the subspace angles are computed as:

$$\tilde{A}_x^T \tilde{A}_y = U\Sigma V^T \tag{62}$$

$$\theta_i = \arccos(\sigma_i) \tag{63}$$

Where $\sigma_i$ is the $i$-th singular value defined by $\Sigma$. $\theta_i$ is defined as the $i$-th principal angle. We report the maximum principal angle between two dynamics operators, using the scipy.linalg.subspace_angles function.

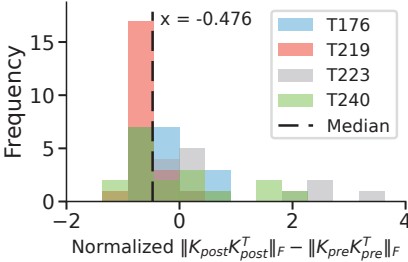

Figure 23: Normalized change in controllability energy across the nTc $(||K_{post}K_{post}^T||_F - ||K_{pre}K_{pre}^T||_F)/||K_{pre}K_{pre}^T||_F$ aggregated across four rats.

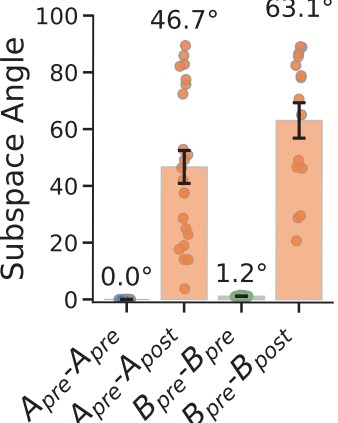

Figure 24: Subspace angles of the input and state operators within and between time periods for rat T223. $A_{pre} - A_{pre}$ (likewise $B_{pre}$) denotes the noise floor via split-halves comparison.

