# OpenReview forum: "InputDSA: Demixing, then comparing recurrent and externally driven dynamics"
_ICLR.cc/2026/Conference — ICLR 2026 Poster_

### Official Review · Reviewer_GeRu · 2025-10-20

**Soundness:** 4
**Presentation:** 3
**Contribution:** 4
**Rating:** 6
**Confidence:** 4

**Summary:**

This work proposes InputDSA, an extension of DSA which considers inputs when defining a distance measure between data produced by two dynamical systems. To accomplish this, subspace DMDc is proposed which leverages results from prior works (N4SID) for subspace identification. A variety of synthetic and real data experiments are considered to show the utility of InputDSA.

**Strengths:**

- The text is overall easy to follow and well-written.
- Does a good job at pointing out a technical gap, that DSA and its extensions have not explicitly considered differences in the effect of external inputs.
- The experimentation is very diverse and does a good job at demonstrating practical use cases for computational neuroscience.
- Good description of hyperparameters in the appendix.

**Weaknesses:**

- The baselines seem insufficient in experiments 3.2, 4.1, and 4.2 since they primarily compare against only DSA which does not learn an input operator. As a result, it is unclear whether the practical performance improvements for each scenario is due to the proposed SubspaceDMDc or due to explicitly considering input in the metric formulation or both.
- Synthetic experiments do not consider process noise (noise over time added to the latent state) which is an important characteristic of neural signals due to their intrinsic stochasticity. Could the authors provide experiments that clarify whether InputDSA is also robust to increasing levels of process noise or not? A simple example would be experiment 3.1 where we also consider $x_{t+1} = A(x_t + gF {\rm tanh} (x_t)) + B(u_t + {\rm tanh}(u_t)) + \nu_t$ where $\nu_t \sim \mathcal{N}(0,\sigma^2)$.


Minor
- Naming the approach “InputDSA” (used in Fig 2 caption) and the distance between input matrices $B$ as “input DSA” (second term in eq. 6 used in Fig 2d,e, and f title) may cause some confusion.
- $\alpha$ is used twice which may cause some confusion. In line 437 as the exponent of the power law, and in equation 6 as a balancing coefficient.

**Questions:**

- In Figure 2d, the scale between input DSA for DMDc and SubspaceDMDc is very different. How do we determine what magnitude of effect is required to conclude that there is a difference between two systems?
- How does the distance metric change for each experiment for different values of $\alpha$? Is this an important hyperparameter to tune or can we always use $\alpha=0.5$? When would we set $\alpha \neq 0.5$?
- How does the window size affect the InputDSA distance metric? Does window size need to be carefully selected to get reliable results?
- How do you select the rank of the latent space? Is the InputDSA still reliable when there is a mismatch between true and estimated rank?
- Is the InputDSA metric differentiable? What is does the computational cost of computing InputDSA scale with larger latent spaces and longer time series?
- In experiment 3.2, is the inputDSA metric also robust to phase changes of the input?
- is it always the case that we want to consider rotated state dynamics as similar?

---

> ### Author Response · Authors · 2025-11-19
> **Weaknesses addressed, thanks for the helpful feedback**
>
> We thank the reviewer for their positive feedback and appreciation of our methodology and results. We also appreciate the critiques, and have made modifications accordingly:
>
> ## Weakness 1
>
> We've done baseline comparisons with DMDc in both experiments (Figures 4 and 5), which we've added to supplement (**Fig. 17 and 19, in Section P.1 and Q.1 respectively**). It seems that the analysis of Figure 4 with DMDc gives some similar results -- this is expected given that the states in this dataset are fully observed, given that they are simulations. However, because these simulations are noisy, they do not estimate the state similarities as robustly, failing to compute the same ordering as SubspaceDMDc. In the neural dataset (Fig 5), however, DMDc fully washes out the result and cannot identify any differences between the two  time windows in the task. Together, this suggests that DMDc is viable in the cases of full observations with minimal noise, as we have argued earlier in the paper (and indeed, Fig 2F top suggests that when the system is fully observed, DMDc is applicable).
>
> ## Weakness 2
>
> We've run a rigorous analysis on the effect of process noise **(Fig. 10, Appendix Section K)**. We reran the analysis in 3.1 with random gaussian noise added to the state dynamics, and measured the silhouette score of both the state and input distance matrices. We find that both are robust -- the state silhouette score remains above 0.8 for all levels of process noise, while the input silhouette score gracefully degrades from 0.75 to 0.4. At the largest levels of noise, Systems 1 and 3 are still related as similar in the input, but systems 2 and 4 are not.
>
> ## Minor weakness 1
> We agree about this confusion -- we have changed inputDSA and stateDSA in Fig. 2 to "input distance" and "state distance" respectively.
>
> ## Minor Weakness 2
> We have changed the expression of the metric in **Section 2.2** to reflect how it is truly computed -- this removes one instance of $\alpha$. Please see the overall comment above and the new draft for further details. For the power law fit in Fig. 5D, we have changed the exponent variable name to $\beta$, which should reduce confusion as well.
>
> ## Questions
>
> 1.  The magnitude of the effect required typically depends on the size of the fitted dynamic operators, metric used, and structure in the data. We compare to a baseline set of scores via a split-halves analysis, in which we measure the within-system distribution of similarity scores. This can be done either by (1) subsampling the data within a single dataset, or by (2) running multiple simulations with the same model. The resulting within-system scores provide an empirical noise floor against which cross-system comparisons can be evaluated. For example, in Fig 2. C, D, we run 4 examples of the same model with different initial conditions, inputs, and noise, which can constitute our baseline. **In Fig. 11 in Appendix L**, we ran a more rigorous analysis by computing the within-system scores via both the aforementioned approach (1) and (2), and compared them to the cross-system scores. The within-system distances computed via (1) and (2) are typically similar, and both are markedly smaller than the cross-system distances.
>
> 2. As mentioned above, we've chosen to move away from the $\alpha$ parameterization of the metric, and instead focus on the key metrics that are interpretable: joint inputDSA (total score involving both $A$ and $B$, essentially $\alpha =0.5$), jointly optimized state score (the score on $A$ with the learned $C$ using both $A$ and $B$), jointly optimized input score (the score on $B$ with jointly learned $C$), separately optimized state score ($A$ score, with $C$ learned just on $A$--essentially $\alpha = 1$), and separately optimized input score ($B$ score learned just on $B$--essentially $\alpha=0$).
>
> 3. We apologize for not adding sufficient detail on hyperparameter tuning in the main body. We've added a paragraph to **Section 2.3, Line 213-218** entitled **Tuning SubspaceDMDc for InputDSA** detailing this, please check it out. We hope answers your question on hyperparameter tuning.  Furthermore, **in Appendix Figure 8, Section J.1**, we test InputDSA’s robustness to changes in hyperparameters using partially observable nonlinear systems. We found that the InputDSA score is indeed highly robust to the choice of delay and rank, even when the chosen rank does not match the system's true rank.
>
> 4. yes, the metric is differentiable--the gradient would be taken post-optimization. The metric itself uses Procrustes optimization, which scales as $O(n^3)$ in the state size due to the computation of an SVD. The SubspaceDMDc model complexity scales approximately as $O(h^3)$ where $h$ is the dimension of the Hankel matrix: $d*n$, where d is the number of delays in the delay embedding. However, there exist randomized SVD methods that can bring this scaling down to $O(n^2)$ and $O(h^2)$.

---

> ### Author Response · Authors · 2025-11-19
> **Questions addressed (cont.)**
>
> 5. In **Figure 14, Appendix Section M.1**, we examined the effect of time-shifting the input (this was our interpretation of 'phase change', please correct if it's wrong). Using partially observed nonlinear systems, we computed the correlation between the InputDSA distance matrix estimated with the ground-truth input and the matrix estimated using time-shifted versions of that input. The similarity structure of the input-driven dynamics is highly robust to temporal shifts, maintaining a distance correlation of approximately 0.75 even after a 20-step shift. The estimated intrinsic similarity structure is less stable, while still yields a distance correlation around 0.5 across shifts of up to 20 steps.
>
> 6. No, if you have a meaningful or aligned basis between systems, you might not want to. We think about this similarly to how shape metrics are used in neuroscience--when each system either is randomly partially observed, or there is no direct / clear alignment between dynamics, then the observation could be subject to random rotations. If the basis is already aligned, then the optimization method could be removed.
>
> ___
> Thanks again for the suggestions, we believe the manuscript is much more complete as a result of these additional baselines and analyses.

---

### Official Review · Reviewer_kqZq · 2025-10-29

**Soundness:** 4
**Presentation:** 3
**Contribution:** 3
**Rating:** 8
**Confidence:** 3

**Summary:**

The paper proposes InputDSA (iDSA), extending Dynamical Similarity Analysis (DSA) to non-autonomous systems by explicitly modeling both (i) intrinsic/recurrent dynamics and (ii) input-to-state mappings. Estimation uses a new Subspace DMDc (a subspace-identification variant of DMD with control) to mitigate the partial-observation bias that makes vanilla DMDc fail. Applications include synthetic systems, RL-trained RNN agents, and neural data show shifts from input-driven to intrinsic dynamics.

**Strengths:**

1. This paper solves a clear problem gap - ‘Most dynamics-similarity methods ignore inputs.’
2. The proposed method has partial-observation robustness. Subspace DMDc directly addresses the bias where inputs instantaneously affect observed states and, via unobserved states, which is an excellent point for neural data.

**Weaknesses:**

1. Since InputDSA mixes intrinsic and input dynamics (via alpha in equation 6), it would be useful to show how sensitive results are to alpha. In other words, how to choose alpha when applying this method?
2. The paper should clarify how the delay parameter d is determined or selected in practice.
3. The results lack statistical uncertainty estimates on results, such as statistical uncertainty test on silhouette scores.

**Questions:**

Please see weakness.

---

> ### Author Response · Authors · 2025-11-19
> **Weaknesses addressed, thanks to the reviewer for helpful suggestions**
>
> We thank the reviewer for their positive feedback and appreciate their suggestions. We're happy that the importance of the problem and viability of our solution was recognized.
>
> ## Weakness 1
>
> When thinking about the value of alpha, we only studied 3 values: (1) $\alpha = 1$ which only compares the state operators, independent of the input (state score, most similar to the original DSA). (2) $\alpha = 0$, which only looks at the geometry of the read-in operator $B$, and how different components of the input are rescaled / mixed together. (3) $\alpha=0.5$, which looks at the interaction of intrinsic and inputs and is most similar to studying the differences in controllability, which is a much more interpretable measure.
>
> However, we've chosen to move away from the $\alpha$-based formulation of the metric in favor of the joint comparison metric that more directly studies controllability. We replaced much of **Section 2.2**. in the paper with new detail describing this metric, and would greatly appreciate the reviewer reading this revised methodological introduction as well as the overall comment.
>
> In practice, this doesn't change our results (which we computed using the controllability metric anyway, as mentioned in the appendix). $\alpha = 0.5$ corresponds to looking at the joint score (as in Fig. 6 in the appendix), $\alpha = 0.0$ is the same, looking at the state DSA score. In practice, we found it most informative to look at the jointly-optimized input score (i.e. using the C from the $\alpha=0.5$ optimization and just studying the distance between input operators).
>
> ## Weakness 2
>
> We agree and have added a paragraph to **Section 2.2, Line 213-218** on tuning SubspaceDMDc: please see the section entitled "Tuning SubspaceDMDc for InputDSA"
>
> ## Weakness 3
>
> We added a statistical test on the silhouette scores in **Fig. 2F** to assess that the silhouette scores are significantly greater in SubspaceDMDc versus DMDc and DMD. Specifically, we ran a one-side Mann Whitney U-test between the distribution of silhouette scores of \{DMDc or DMD\} and SubspaceDMDc for each system dimension in Fig 3F. SubspaceDMDc had significantly greater scores in every condition except three state scores (at dimensions 8, 200 and 500 respectively). We also added statistical tests on **Fig 5** for each subpanel.
>
> ___
>
> Thank you again for the positive feedback.

---

### Official Review · Reviewer_R5Xx · 2025-10-31

**Soundness:** 3
**Presentation:** 4
**Contribution:** 3
**Rating:** 6
**Confidence:** 4

**Summary:**

The paper proposed a new method, inputDSA, to compare between observed trajectories of two dynamical systems. It is an extension to DSA that addresses the important point of external input. The intuition behind both methods is that lifting the observations to a highD space can make the dynamics linear (which is correct for infinite dimensions via Koopman theory). Comparison in the linear domain is relatively easy. Because neuroscience applications typically have external input, and this can completely bias the comparison process, the current work shows how to incorporate inputs into the framework. This has the added benefit of making the algorithm more efficient. The authors demonstrate the method on several synthetic examples, and on experimental neural data.

**Strengths:**

This is an important problem. With the availability of highD neural recordings, inference of dynamical systems from data is increasingly done using many methods. DSA is more of a comparison then direct inference, but it contributes to this field. External input is a crucial aspect of many of these methods.

The paper is clearly written, explains the problem, the approach and the results.

The focus on partial observation is highly relevant to possible applications, and done systematically.

**Weaknesses:**

The interaction between A and B are almost not discussed. Appendix J touches upon it, but the results in the main text mostly show input similarity and state similarity. For instance, in the neural data example, Figure 5F shows angles within A or B. But an interesting question is whether the input arrives to directions that “matter” to the intrinsic dynamics. Can that be measured using eigenvectors of A and their relation to B?

The limited effect of wrong inputs on state estimation. Results such as Figure 3D show that getting the input completely wrong has little effect on the quality of intrinsic (A) estimation. This seems somewhat contradictory to the original motivation of the paper.

**Questions:**

The scope of applicability is not clearly defined. When should this fail? What are the conditions (reachability, full rank) – and how can you verify this. For instance, how do you know it works for your examples?

Eq 1. This is not a symmetric measure. For instance, the dimensions can be different. Might be useful to discuss what happens if internal/input dimensions differ between systems.

Eq 10 – what is the motivation for this form of dynamics?

What is the meaning of the numerical value of scores? The original DSA paper used embedding in a lowD space, whereas you mostly use silhouette. What is the motivation for this choice?

Fig 2F – Partial observability degrades state estimation (top panel). From Figure 2A, one can think that having access to the true input is a way to rescue this estimation, because we have indirect information of the unobserved units. There is a small increase in the state silhouette score, but I wonder how much can be rescued. For instance, if the input is more than 1-dimensional – does this help?
No reference to Fig 3AB from main text

Fig 3D – perhaps use bars, violin, or anything else that does not connect the dots, when the X axis is categorical.

Fig 3D – why is the intrinsic estimation unaffected by wrong (random) inputs? Are there inputs (e.g. anti-correlated) that will hurt the intrinsic correlation?

Line 338 typo? “crucial to how inputs”

Plume tracking: “This implies that inputs excite more dimensions of the RNN in Top agents (Fig. 4E), thereby allowing them to more effectively incorporate recent information to inform action selection in the future.” – I’m not sure I follow the logic here. Why should high-D input be more effective in incorporating information? Shouldn’t this be linked to the relation between these directions and those of the intrinsic dynamics? If these higher dimensions are orthogonal to internal dynamics, for instance, then it should just be noise.

Neural data: Aligning to the ntc point. If I understand correctly, this uses the result from a different pipeline (identifying this point), and then does a PSTH using this shift. Perhaps worth emphasizing.

Neural activity: why the need to use PCA and isomap for preprocessing? Does using Isomap change any assumptions about the dynamical system? For instance, it could be that the discarded dimensions are important for the closure of the dynamics.

Line 436: power law fit. Dynamics is usually dominated by the slowest mode. Why use this method of fitting? While several slow modes can interact, the very fast modes are not expected to be relevant here – but they are used in the fit. Also – integration (pre) is expected to have line attractor like dynamics, or very slow intrinsics dynamics. Isn’t the result opposite to this intuition? If so, do you know why?

Line 442: reflected – reflecting

Appendix K tables are in the next page, which is confusing.

---

> ### Author Response · Authors · 2025-11-19
> **All questions and weaknesses addressed in response, thanks for helpful additions to the manuscript**
>
> We appreciate the reviewer's recognition of the importance of the problem we seek to tackle, and are glad that they think the paper is clearly presented. We appreciate that the reviewer thinks our analysis of partial observability and noise robustness are systematic, and they will be happy to hear that we have added further robustness tests and statistical quantification.
>
> ## Weakness 1
>
> The first point in the weaknesses is an excellent one -- comparison of an input controlled linear system is not complete without understanding the interactions between A and B. We agree that this is highly important! InputDSA handles this situation explicitly and we apologize for not being clearer. We have expanded our description of the inputDSA method to better highlight controllability. Instead of keeping the formulation of equation (6) with the $\alpha$ (which only implicitly measures the interactions via joint optimization of the two terms), we chose to instead highlight the controllability metric, which is how the metric was being computed in the first place. It's also worth noting that this is more correct anyway, as the two metrics (equation 6 in the first submission versus 45 in the supplement) are not entirely equivalent, but have the same zeros. We therefore rewrote **Section 2.2** to follow this detail. We have uploaded a new version of the manuscript, please take a look. We've correspondingly updated figure 1. Under this framing, the joint score *explicitly* measures the differences in the geometry of controllability of the two systems, from which the state and input matrix can be compared as well.
>
> We also agree that Figure 5 could analyze controllability directly, rather than looking at A and B separately. To that end, we computed the eigenvalues of the controllability Gramians, and studied how their distributions changed pre and post the neural time of commitment (nTc). We found that the overall power of the Gramians, as measured by the Frobenius norm (which captures the sum of squared eigenvalues), significantly decayed post-nTc (**Fig 22 and 23 in Section Q.3**, p<<0.05 for all animals, Mann-Whitney U-test, median change of -46\% in controllability energy). A larger eigenvalue in the Gramian indicates that less input is required to control the system in the corresponding direction, hence this confirms our prior analysis that controllability decreases across the nTc. We have added this analysis into **Figure 5F** and moved the subspace angle analysis (previous Fig 5F) to Appendix for space **Figure 24, Appendix Q.4**.
>
> ## Weakness 2
>
> The second weakness is well-taken, but is worth clarifying: state similarity scores are robust to not knowing the true input, but knowing the true input (or a decent surrogate) can serve to sharpen them. This is corroborated with Figure 2, in which the state score fit with DMD (no input) still reveals the same clustering (as does DMDc, even though there is the problem of partial observability that alters the input score). We think this result makes sense and also adds further justification to the success of the original DSA method, as it has previously been applied to input-driven systems (even in the original paper, Ostrow et al. 2023). The added benefit of inputDSA is therefore: (a) sharpening the state score -- Fig 2C, and (b) adding new comparison metrics that disentangle the input score and study the interactions between input and state.
>
> However, we did find it odd that in Fig 3D, the input score correlation with ground truth was still strong even with random inputs -- unlike the state score, we would expect this to be close to 0 correlation. Between submission and release of reviews we conducted experiments in between the submission and release of reviews to assess this problem. We identified a bug in comparison, in which the *diagonals of the matrices were also being correlated*. This will systematically inflate correlation as distance matrices always have 0 along their diagonal. In the updated **Fig 3H**, we fixed this issue: the random input now has close to 0 correlation, and for the other surrogate inputs, more highly-correlated surrogates tend to yield more accurate similarities.

---

> ### Author Response · Authors · 2025-11-19
> **Questions addressed here**
>
> 1. Scope of applicability: We briefly touch on the applicability in the final paragraph of the paper (Line 531-535), but apologize for not discussing sooner. As mentioned, standard failure modes of subspace identification hold here -- if an input is not sufficiently rich or persistently exciting, some directions of B may not be estimated appropriately, or have high variance. When it comes to comparison, however, perfect approximation is not necessary--sufficiently accurate fits can yield useful comparisons, especially when the goal is to find an ordering (e.g., BrainScore, Schrimpf et al. 2020, where models are ranked based on fits to data). Methods that compare the geometry of neural activations such as Linear Regression, Representational Similarity Analysis, Singular Vector Canonical Correlation Analysis, Centered Kernel Alignment, and Procrustes Analysis (Schrimpf et al. 2020, Kriegeskorte et al. 2008, Raghu et al. 2017, Williams et al. 2021), share a similar philosophy: it is not possible to characterize the full structure, but the approximation from the given data is often sufficient to yield scientific insight. Empirically, Figure 8 in the supplement demonstrates this phenomenon: on a 20-dimensional latent system (with only 2 dimensions observed), SubspaceDMDc identifies the correct similarity structure with as few as 9 modes in the system.
>
> 2. Eq 1. Indeed, the conjugacy map $\phi$ need not be invertible (a semi-conjugacy, which we believe is what the reviewer is referring to?). Two systems with different extrinsic dimensions can be conjugate if they have the *same intrinsic dimension*. Even if not perfectly conjugate, this could be effectively realized in data if there is a separation of timescales between some dimensions, due to the presence of quickly decaying directions. If the rank of a DMD model is overestimated, this is typically what will emerge (also known as a spurious mode). In practice, these can be filtered out by tuning. However, if two systems have different dimensions, one could (a) overestimate the smaller system, potentially adding some noise to the metric, or (b) underestimate the larger system, which could bias the estimates. In practice, we've found that the choice depends on the eigenstructure of the smaller modes (which are truncated first). If the noise is small, overestimation is likely safer (and these modes can also be zeroed out with an L1 regularization).
>
> 3. Eq10. This is a random RNN -- the SubspaceDMDc can naturally fit linear systems so we wanted to add to demonstrate its capabilities beyond linear systems, by adding nonlinear structure to both the input and the state functions. **We've correspondingly renamed it in line 225**.
>
> 4. Embedding in a low-D space is simply a way of visualizing separation between classes, which contains the same information as visualizing the distance matrices themselves--they're complementary. Decoding of classes from the distance matrices or computing the silhouette score are highly complementary as well. The DSA paper compared only 2 metrics (Procrustes vs DSA) on many classes, which makes embedding more visually appealing. In our Fig. 2, we have 5 similarity matrices, and we thought showing 5 scatterplots may be more visually cluttered and harder to interpret at a quick glance.
>
> 5. Fig. 2F We apologize for the lack of clarity--the input in this analysis used in inputDSA is the true input, only the observed fraction of the state is being modulated. We agree that it would be interesting to assess scaling in input dimensionality as well, though. In Appendix Figure 9, we vary the dimensionality of the low-pass filtered white noise input to partially observable nonlinear systems, and evaluate the Silhouette score (computed against ground-truth labels) at different levels of partial observability. As shown in **Fig. 9 in Appendix J.2**, increasing the dimensionality of the input provides only a very modest improvement in recovering the true state similarity structure, while causing a slight reduction in the identifiability of the input-driven dynamics. We think this is because the 1D low-pass filtered white noise is already persistently exciting, which provides sufficiently rich information to robustly identify the dynamic operators, therefore any additional dimension minimally improves system identification.
>
> 6. We've added references to Figure 3 A and B in the introduction to the analysis done in this section. Thanks for the catch.
>
> 7. Fig 3D - We apologize and have reworked this visualization. **They are now bars in Fig. 3H with colors corresponding to those in Fig. 3G for clearer visual connection between subplots.**

---

> > ### Author Response · Authors · 2025-11-19
> > **Questions answered (continued)**
> >
> > 8. Fig 3D - We appreciate this catch and were confused about it ourselves-- see above section (Weakness 2) for a discussion. Regarding the second question, a strongly correlated input could in theory bias the regression (also termed collinearity). However, the preprocessing step of SVD whitening generally reduces this effect. The worst case setting would be when the input is a linear function of the state (as in a closed loop feedback control system). However, this is a rare case for nonlinear systems, but there also exist methods for subspace identification in these settings that could be applied (see Van der Veen et al., 2013 "Closed-loop subspace identification methods: an overview" for examples).
> >
> > 9. Typo - Thank you for catching this. We meant to write "it is crucial to study how inputs interact ..." we have updated this in the manuscript.
> >
> > 10. Plume tracking - We apologize for not clarifying: the state operators learned by subspaceDMDc are full rank -- therefore, the higher dimensions will not be orthogonal to the internal dynamics. *From a controllability perspective, a more slowly decaying singular value spectrum of B means that more directions are more directly controllable by the input, rather than indirectly controllable through the input's action on other states. Direct controllability implies that these directions in state space can be moved immediately, rather than being filtered with a delay through other states*--hence our claim. We have added detail of the discussion of into Section 4.1 at **Line 419-420**. Please let us know if you have any further questions.
> >
> > 11. Neural data - it is true that the ntc was computed via a different pipeline. We chose this for the sake of computational efficiency and visual clarity. In practice, one could compute these metrics across a sliding window of time in the trial, which would give a trajectory of distances (compared to the first window, for example), and operator magnitudes / eigenvalue exponents. One would potentially see either a smooth change or a discrete step (smoothed if there are overlapping windows), which could give further information about how the dynamics change. We additionally note that, for each session, Subspace DMDc models were fit to the pre-nTc and post-nTc neural activity (and input) using all preprocessed single-trial data, without applying trial-averaged PSTH estimation before model fitting. By “aligning to the nTc,” we simply mean that we split the single-trial neural and input time series at the time point of nTc (with further preprocessing to ensure equal segment lengths and to avoid extending into the movement period). We have updated the manuscript on **Line 439-440** to make it explicit and avoid confusion.
> >
> > 12.  Neural activity - processing with PCA and Isomap serve to create better fits with SubspaceDMDc, and therefore provide more robust comparison scores (in fact the preprocessing steps can be thought of as creating a nonlinear feature space for SubspaceDMDc). While the discarded dimensions could be important for closure of the dynamics, they are often discarded due to lack of sufficient sampling to robustly estimate DMD modes in these directions. This means they may add more noise than signal to the comparison. We've added detail on fitting SubspaceDMDc with the motivation of identifying a space that can best approximate the dynamics as linear (see overall comment).

---

> > > ### Author Response · Authors · 2025-11-19
> > > **Questions answered (fin)**
> > >
> > > 13. We utilize a power-law fit to quantify the distribution of eigenvalues, which is informative about the system as a whole, similar to how random matrix theory computes distributions of eigenvalues. We agree that dynamics are dominated by the slowest modes, but fast and slow modes can still interact in non-trivial ways, especially when the system is non-normal. As indicated by our hyperparameter sweep for this analysis (Appendix Fig. 20, Section Q.1), a rank 6 model was best fitting in terms of both MASE and AIC scores, so it appears that the are still relevant to the dynamics. However, we agree that this may not be the most direct way to quantify the change in controllability. To do so, we compute the eigenvalues of the controllability matrices for each dataset, which now more directly encodes the interaction between input and recurrence we care about. We find that each of the eigenvalues diminishes across NTC, implying that every direction of the dynamics becomes less input-controllable.  We decided to add this into the main paper (**Fig 5F**, further detail appendix **Figure 22 and 23 in Section Q.3**) since we think it highlights the main point. Regarding the line-attractor point, Luo et al (2025) find that the integration dynamics in this dataset are not best described by a line attractor (Fig 1i in their paper, flow field visualization has no slow direction). Regarding the speed of dynamics during integration, our result is not necessarily opposite -- the first three eigenvalues are all close to 1, indicating that they capture slow modes of the dynamics. Because integration in this dataset is nonlinear, it may require multiple modes in SubspaceDMDc to adequately represent. Additionally, even though the top 3 eigenvalues remain roughly at the same value, they describe different directions in neural activity space (Fig 5F, subspace angle between pre- and post- operators, *now appendix figure 24*).
> > >
> > > 14. reflecting: thank you again for catching the typo
> > > 15. Apologies - we've reorganized the layout of the appendix so that the pseudocode directly follows the section label.
> > > ___
> > > Thank you again for all the useful suggestions, we believe the manuscript is significantly better as a result.

---

> > > > ### Comment · Reviewer_R5Xx · 2025-11-27
> > > > **Post rebuttal**
> > > >
> > > > I thank the reviewers for the thorough rebuttal and modifications. Many of my concerns have been answered and I am raising my score to 8.

---

### Official Review · Reviewer_3e6A · 2025-11-01

**Soundness:** 3
**Presentation:** 2
**Contribution:** 2
**Rating:** 6
**Confidence:** 4

**Summary:**

This paper introduces InputDSA, which aims to extend Dynamical Similarity Analysis (DSA) to systems with external inputs. The method fits, for each system, a linear controlled latent state-space model h_{t+1}=Ah_t+Bu_t via a subspace ID-style procedure, and then defines a similarity metric that compares the recurrent term A (“intrinsic dynamics”) and the input term B (“input-driven dynamics”) across systems after alignment. The problem is important, and the core idea, explicitly separating internal dynamics from input drive and comparing both, is novel and interesting.

**Strengths:**

The synthetic experiments are well designed and give evidence that under good excitation conditions and with partial observability, the proposed estimator can recover similar recurrent structure vs similar input structure. The applications to trained agents and to neural population data are very appealing and show how the method could validate and generate hypotheses about “what part of the dynamics actually matters.”

**Weaknesses:**

There are some serious concerns with the clarity and interpretability of the current manuscript:

1.	The main text never clearly defines what state is being modeled. In practice, the method learns a latent linear model (A,B) on top of lifted observations via Subspace DMDc. But the paper (especially schematic Fig. 1 and Sec. 2) blurs together true system state, observed activity, and this identified latent state, and then talks about A and B as if they were the system’s real intrinsic vs input-driven dynamics. As written, it’s not clear what is actually being compared from the main text alone.

2.	Many essential details of the identification procedure (how A,B are estimated, model order selection, assumptions needed for demixing) are only in the appendix. From the main body, it is difficult to understand the method and pipeline.

3.	More important: the application to neural data is not well justified, considering the fact that inputs in neural tasks are usually unknown/partial. In real neural recordings, the “inputs” we can measure are usually low-dimensional task variables, not persistently exciting signals. The paper acknowledges this and presents a robustness test with surrogate/noisy inputs in Sec. 3.2, and that test is done in an RNN control setting with richer continuous inputs. However, moving to neural data in Sec. 4.2 and the claimed interpretation “the circuit becomes intrinsically driven after commitment, whether the A/B split is still meaningful becomes unclear since the inputs here are pretty abstract and simple."

Overall, I think the idea of this paper is very interesting, and the paper brings some valuable contributions. But the paper in its current form oversells its interpretability on neural data, and the core method is under-specified in the main text. I would like to see a revision that: (i) clearly defines the latent state and is explicit that InputDSA compares identified models; (ii) makes the system ID procedure understandable without digging too much through the appendix; and (iii) tones down the neural interpretation in light of partial/weak inputs.

**Questions:**

See Weaknesses above.

---

> ### Author Response · Authors · 2025-11-19
> **All weaknesses addressed; thanks for the helpful feedback (Weaknesses 1+2 here)**
>
> We appreciate the reviewer's positivity about inputDSA and are grateful for the constructive feedback. Your concerns are recognized; we'll clarify some confusions and discuss how we've modified the manuscript to account for them.
>
> ## Weakness 1
>
> We apologize for the confusion--the notion of state in the setting of the Dynamic Mode Decomposition (DMD) is somewhat abstract, as the primary goal of this methodology is to find an embedding of the data that is conducive to linear modeling of the dynamics, i.e. some $\phi$ such that $\phi(x_t) \approx A\phi(x_t)$ for some A. The DMD seeks to approximate the Koopman Operator (Koopman, 1931), which is an infinite dimensional linear operator that always exists for dynamical systems. Operators of dynamically similar (conjugate) systems have well defined relationships, which motivates DSA, and inputDSA in turn.
>
> With this formulation in mind, the notion of state is not necessarily interpretable at face value (although the DMD eigenmodes can be used to interpret the dynamics of the original state space). For example, the Hankel Alternative View of Koopman (HAVOK, Brunton et al. 2017, which DSA leverages) utilizes delay embeddings as a nonlinearity, which can help resolve partial observation, just as in subspaceDMDc (the HAVOK eigenmodes on the Lorenz attractor are interpretable, and relate to almost-invariant sets of the system--Fig 6 in their paper). From Takens' delay embedding theorem, delay embeddings reconstruct partially observed systems *up to a conjugacy mapping*, which can potentially be complex. Another example is the Extended DMD (Williams et al., 2015), which uses kernel regression--in this setting the 'state' is the RKHS of the kernel one utilizes. The state of a DMD model is therefore some abstract mapping on a dynamical systems' state.
>
> With this background in mind, what we're trying to compare is a finite-dimensional approximation of the Koopman operator. **We have added clarification text in section 2.3 (Line 168-173) that this is the goal of subspaceDMDc, to highlight the distinction that we're not doing latent state identification in the standard way in computational neuroscience or ML** (this is usually a dimensionality reduction, whereas we tend to perform a dimensionality expansion). **We have also added references to the schematic steps in Figure 1 in sections 2.2 and 2.3, to highlight what is being compare and how it is being computed.**
>
> ## Weakness 2
>
> We apologize for not including sufficient details of the identification procedure in the main body. The technical details of subspaceDMDc are complex, but we agree that understanding how the method works in practice is important. We have added the following details in section 2.3 in response to your raised points:
>
> **System ID procedure for estimating A and B SubspaceDMDc**
> We have added the intuition behind SubspaceDMDc under the "Issues of Partial Observation" subsection **(Line 206-211)**: *"The algorithmic idea behind subspace identification is similar to that of instrumental variable regression: the lifted state data to-be-predicted (future data $y_{t+1}$) is projected onto the basis of the past input and lifted state data ($y_t,u_t$) before estimating a $A$ and $B$ via linear regression. This has the added benefit of projecting out observation and process noise, thereby providing noise robustness (Verhaegen,1994, Verhaegen and Verdult, 2007). For technical details on the subspace identification algorithm, see Appendix E. "*
>
> **Model order selection**
> We have added a short section after "Issues of Partial Observation" to address how hyperparameter tuning for InputDSA can be done **(Line 213-218)**:
> *{**Tuning SubspaceDMDc for InputDSA:**     SubspaceDMDc has three key hyperparameters: the rank of the linear operator, the number of delays used in subspace identification, and the type of nonlinear basis used. In practice, the most general method we use to tune SubspaceDMDc is to pick the smallest rank model that best predict future states of the system. Because SubspaceDMDc must infer the latent state from prior data, we utilize Kalman Filtering (Kalman,1963) for efficient next-step prediction. This is effectively done with the Akaike Information Criterion (AIC) along with other metrics we discuss in the appendix."*
>
> **Assumptions behind demixing**
> We generally assume that inputs are either known or a useful surrogate input can be constructed that has similar information content as the true input.  This allows the estimation of B separately from A. It is still an open question as to how to infer inputs to a dynamical system if none are given (E.g. Pandarinath et al., 2018). Note that DMDc (Proctor et al., 2016) assumes known input as well. We've explicitly mentioned now this on **line 187**.

---

> ### Author Response · Authors · 2025-11-19
> **All weaknesses addressed; thanks for the helpful feedback (Weakness 3 here)**
>
> ## Weakness 3
> This is an excellent question and we'll clarify: To your first point on the surrogate input, it is true that the inputs are not the same as the real inputs into the brain regions we observe. The ability of the model to robustly capture the structure of B is indeed dependent on the richness of the input. In this section, however, we followed prior modeling work in designing the input (Luo et al., Nature 2025), which modeled the data via a neural network:
>
> *"To reconstruct neural activity, FINDR uses a deep neural network G that takes the spike trains of N simultaneously recorded neurons y and the sensory click inputs u in a given trial to obtain the time derivative of the d-dimensional latent decision variable.
> Here, T is the number of time steps in a given trial, ut is a two-dimensional vector representing the number of left and right clicks played in a time step"* (Luo et al., Methods section "FINDR", describing equation 8).
>
> It is reasonable to suggest that the surrogate inputs in the neural data analysis are abstract and simple (as described above). To assess whether these inputs were rich enough to robustly model the input operator, we computed the standard measure of persistent excitation used in control theory to answer this question. An input is persistently exciting if the matrix
>
> $M(t) = \int_0^t \tilde{u}(t)\tilde{u}(t)^Tdt$
>
> is positive definite, i.e., the covariance matrix of the embedded input is full rank. In this analysis, we used models with 5 delays and rank 6 (neural states were 3 dimensional and inputs were 2 dimensional), so this covariance matrix needs to be at least rank 6. We computed this using the delay-embedded input as $\tilde{u}$ for each session of all four rats, 80 in total with 100 trials on average per session. In each session, the 6th singular value of this covariance matrix is much greater than 0 for each session, with the bottom 10\% quantile having value of 0.22 and the bottom 1% quantile being 0.01. This suggests that even the input is simple, it is persistently exciting enough to robustly identify a useful B operator, given that there are a large number of trials. **We've presented a sample visualization of input for one trial and reported the distribution of ranks and last singular values of the covariance matrix in Figure 21 in Appendix Section Q.**
>
> Likewise, we agree that the neural interpretation could be toned down--based on our results, it is more accurate to say that \textbf{the system becomes less controllable over time}, rather than autonomous. We've therefore added the following text at **Line 480-484**:
>
> *We found that the post-commitment periods consistently showed smaller $\beta$... (we have changed the power-law exponent to $\beta$ to avoid repetitive use of $\alpha$; new text next) This is directly related to the controllability of the dynamical system, which describes how easy it is for an input sequence to drive the system to arbitrary points in state space (Luenberger, 1979). Smaller DMD eigenvalues implies
> greater input controllability, which would be expected for a more input-driven system as Luo et al. (2025) identified is the case in the pre-nTc regime.*
>
> We also added an additional analysis directly computing the energy of the controllability Gramian, which measures the 'ease' of controllability (larger implies easier, **Fig 5F and Fig 22, 23 in Appendix Section Q**). Across all animals, the energy significantly decreases after the nTc, suggesting that controllability does indeed decrease.
>
> ___
> We hope these modifications address your concerns and would be happy to continue to revise these sections if they aren't sufficient.

---

### Author Response · Authors · 2025-11-19
**Summary of major modifications, thanks to all the reviewers for helpful suggestions**

Thanks to all the reviewers for their generally positive feedback and critiques. We've made a number of changes to the manuscript that we hope will alleviate most concerns. We've uploaded a new draft with all of our changes, and we'll highlight the main elements here:

## 1. Removing the need for $\alpha$, cleaner and more theoretically-grounded similarity metric
Multiple reviewers asked about studying the interactions between A and B, or relatedly how to pick $\alpha$ (the weighting factor between the state and input similarity metrics). In Appendix section I, we introduced an efficient way to compute the metric, which has a slightly different formulation based on the controllability matrix: $K = [B, AB, A^2B, ... ,A^nB]$  **and removes the need for $\alpha$!**.

The controllability matrix directly encodes the interactions between A and B across time. Comparing this between two systems thereby compares the interaction structure of input and state dynamics. We think that this metric, (which is what is computed after all, Appendix Section I) should take the place of the previous metric (equation 6) in the main paper. **We've also added additional motivation as to why this metric is important in section 2.2, and included detail about how the input and state scores are computed downstream from this** (also in an updated Fig. 1).

To better understand the metric, we added a new theoretical analysis showing that this and the former metric have exactly the same zeros (based on equivalency of input-driven linear systems, **Lemma G.1 in Appendix)**. They are not exactly the same metric, which is why we found it pertinent to switch to the controllability formulation (and we think this is more interpretable as well). Lastly, **we've added a direct controllability analysis to Fig. 5** in which we identified that the neural dynamics become less input controllable across the nTc, for all rats (also see **Appendix Fig. 22 and 23**). We analyzed the controllability Gramians of SubspaceDMDc, which is the matrix that is utilized in the metric.

## 2. More detail on the method's usage in practice

Multiple reviewers asked for more detail about how the method is applied in practice, particularly how to tune the hyperparameters of the DMD models. We relegated most of this detail to the appendix in our submission but recognize its relevance; **we added a paragraph describing the basics of tuning to section 2.3 ("Tuning SubspaceDMDc for InputDSA")** and have performed more tests on the effect of picking different ranks and delays on the results.

In a similar spirit, **we've also added further detail on why this is the goal: the DMD methodology seeks to approximate the Koopman Operator, which is a linear representation of a nonlinear dynamical system with favorable properties for dynamics comparison**. We therefore seek to identify the embedding + model that best describes the dynamics as linear, upon which we can perform comparison. Intuitively, this is the minimal model that best fits the data. This means that the model is not necessarily identifying some meaningful 'state' but rather an abstract state that linearizes the nonlinearity dynamics.

## 3. Further robustness tests and baselines
  We added a sweep of the analysis in Fig. 2 on different numbers of delays and ranks, input dimension, and levels of process noise **(Figures 8,9,10 in the appendix)**, finding that InputDSA is robust here as well.
**We added an additional perturbation analysis in Fig 3** applying random polynomial functions to the input, finding that inputDSA is also robust to these, provided that the Signal-to-Error Ratio (the relative deviation of the perturbation) is not too small. **We fixed the random noise surrogate test on the random target task (Fig 3), so that it now is near 0 correlation, as would be expected** -- the scores were inflated due to comparing the diagonals of the distance matrices as well (which are trivially always 0) and made these visualizations clearer. We ran the analysis of Figs 4 and 5 using DMDc in InputDSA rather than SubspaceDMDc and found that its lack of robustness to noise and partial observation masks multiple of our core results.

## 4. Further detail on failure modes
Multiple reviewers asked about failure modes, which we briefly highlighted in the last paragraph of the paper but have since been elaborated on. In particular, we highlighted: insufficient richness of the input (persistent excitation), limitations on comparison accuracy due to the capabilities of the models to fit the data (garbage in - garbage out).


___
We thank all reviewers for their helpful feedback; we believe the paper is significantly better as a result and we hope they think so too.

---

### Comment · Area_Chair_1rJd · 2025-11-26
**Reminder to Engage!**

Dear Reviewers,

We are one week away from the end of the discussion period and the review responses have been posted. Please read the response and check if the authors have addressed your concerns. Also please acknowledge the review by responding and stating how the response (and updated manuscript if provided) does or does not change your evaluation of the work. Earlier responses allow for meaningful engagement and potential for further clarification.

-Area Chair

---

### Meta-Review · Area_Chair_MHER · 2025-12-04

**Summary:**

The authors introduce InputDSA to measure the similarity of input-driven dynamical systems by demixing intrinsic and input dynamics via Subspace DMDc. Reviewers raised questions that the submission was under-specified regarding the latent state definition and how the metric accounts for controllability and the interaction between A and B. Concerns were also raised about the interpretability on neural data where inputs are unknown or partial, alongside requests for statistical uncertainty estimates and process noise robustness tests.  Additional baselines were also required from reviewers to demonstrate that Subspace DMDc outperforms standard DMDc under partial observation.

**Reviewer Concerns:**

Reviewers raised concerns regarding the theoretical clarity of the method and its robustness under partial observation, specifically questioning the definition of the latent state and the interaction between intrinsic ($A$) and input ($B$) dynamics (e.g., reviewer 3e6A). Several comments focused on the method's validity in neural applications where inputs are often abstract or unknown.

Another concern is that the submission did not explain how the metric accounts for the controllability of the system or how hyperparameters like the weighting factor $\alpha$ and delay embeddings were selected (raised by multiple reviewers). Furthermore, the evaluation rigor was questioned due to a lack of statistical uncertainty estimates and insufficient baselines, particularly the omission of comparisons against standard DMDc and tests involving process noise.

**Reviewer Scores:**

All reviewers are positive regarding the submission, with score 6,6,8,6.

The AC read through the rebuttal and updated manuscript carefully, and is satisfied with the  revision, especially theoretical clarity and ablation studies.

---

### Decision · Program_Chairs · 2026-01-26

Accept (Poster)